# Text-mining uncovers the unique dynamics of socio-economic impacts of the 2018-2022 multi-year drought in Germany

Jan Sodoge[1,2], Christian Kuhlicke[1,2], Miguel D. Mahecha[3,4,5], Mariana Madruga de Brito[1]

[1] Department of Urban and Environmental Sociology, UFZ-Helmholtz Centre for Environmental Research, 04318, Leipzig, Germany
[2] Institute of Environmental Science and Geography, University of Potsdam, 14476, Potsdam-Golm, Germany
[3] Remote Sensing Centre for Earth System Research, Leipzig University, 04103, Leipzig, Germany
[4] Department of Remote Sensing, Helmholtz Centre for Environmental Research—UFZ, Leipzig, Germany
[5] German Centre for Integrative Biodiversity Research (iDiv) Halle-Jena Leipzig, Leipzig, Germany

*Correspondence to*: Jan Sodoge (jan.sodoge@ufz.de)

**Abstract**: Droughts often lead to cross-sectoral and interconnected socio-economic impacts, affecting human well-being, ecosystems, and economic development. Extended drought periods, such as the 2018-2022 event in Germany, amplify these impacts due to temporal carry-over effects. Yet, our understanding of drought impact dynamics during increasingly frequent multi-year drought periods is still in its infancy. In this study, we analyze the socio-economic impacts of the 2018-2022 multi-year drought in Germany and compare them to previous single-year events. Leveraging text-mining

tools, we derive a dataset covering impacts reported by 260 news outlets on agriculture, forestry, livestock, waterways, aquaculture, fire, and social impacts spanning 2000 to 2022. We introduce the concept of drought impact profiles (DIPs) to describe spatio-temporal patterns of the reported co-occurrences of impacts. We employ a clustering algorithm to detect these DIPs and then use sequence mining and statistical tests to analyze spatio-temporal trends. Our results reveal that the 2018-2022 multi-year drought event had distinct impact patterns compared to prior single-year droughts regarding

their spatial extent, impact diversity, and prevalent impact types. For the multi-year drought period, we identify shifts in how impacts have been perceived regionally, especially focusing on legacy and cascading effects on forestry and social activities. Also, we show how regional differences in relevant impacts are controlled by different land-cover types. Our findings enhance the understanding of the dynamic nature of drought impacts, highlighting the potential of text-mining techniques to study drought impact dynamics. The insights gained underscore the need for different strategies in managing

multi-year droughts compared to single-year events.

## 1. Introduction

Droughts challenge human well-being, ecosystems, and economic development worldwide. Their impacts spread across multiple socio-economic sectors such as agriculture, livestock, and waterways navigation (Stahl et al., 2016). They can occur concomitantly (i.e., compounding) or spread from one economic sector to another (i.e., cascading)  (Erian et al.,

2021; de Brito, 2021; Lawrence et al., 2020; Garrick et al., 2018). For instance, drought-related harvest failures in Russia

in 2010, combined with an export ban, led to a global spike in cereal prices. This shortage is assumed to have amplified the food security risk in other countries (Challinor et al., 2018, 2017). Another example is the 2018 summer in Germany, where low soil moisture values caused crop failures, leading to feeding shortages and consequent livestock reductions (de Brito, 2021).

The socio-economic impacts of droughts are not only driven by the biophysical severity of the drought itself but are also shaped by factors such as societal exposure, vulnerability, and adaptation responses (Damian et al., 2023; Blauhut et al., 2015; Lindner et al., 2010; Simpson et al., 2021). Also, impacts influence each other, forming a complex network of cascading and compounding patterns (Chen et al., 2022; de Brito, 2021; Erian et al., 2021). As a result, the socio-economic impacts of droughts are spatiotemporally dynamic and not directly proportional to the biophysical occurrence of drought

hazards. This complexity becomes especially salient during multi-year droughts, which are characterized by prolonged periods of low precipitation and water scarcity, typically leading to regional biophysical feedbacks that exacerbate the hazardous conditions (Miralles et al., 2019). During such events, the effects of vulnerability and exposure tend to build up over time (Kim et al., 2021; De Silva and Kawasaki, 2018). Consequently, the impacts of multi-year droughts are not static; instead, they continuously evolve and change.

The increasing incidence of multi-year drought periods in several regions worldwide (Rakovec et al., 2022; Moravec et al., 2021; Fischer et al., 2021) underscores the need to comprehend how these events impact society. Previous studies have shown that the duration of drought is linked to the emergence of new socio-economic impact types (Yu et al., 2018; Tijdeman et al., 2022; Chen et al., 2022). An intuitive example of the effect of drought duration is the dieback of trees in Australian and California forests due to the extended and intense droughts (Stephenson et al., 2018; Matusick et al., 2018).

Therefore, research on the distinct spatio-temporal impact patterns during multi-year droughts is needed for designing and implementing robust adaptation measures (Liguori et al., 2021; Rakovec et al., 2022).

Over the past years, substantial progress has been made in studying patterns of socio-economic drought impacts (Niggli et al., 2022; Erfurt et al., 2020; Dahlmann et al., 2022; Liguori et al., 2021; de Brito, 2021). However, most of these studies exhibit a limited scope, both spatially and temporally. Their focus on isolated incidents undermines the potential

for broader generalization, leaving uncertainties about the representativeness of the observed patterns across periods not covered by the study. Also, very few studies consider multiple sectors impacted by droughts, and a focus on singular sectors such as agriculture or forestry prevails (for examples of multi-sector assessments, see e.g. Stahl et al., 2016; de Brito et al., 2020; Sodoge et al., 2023). Overall, there is a clear need for a systematic approach that incorporates the multi-sectoral effects of drought during extended periods and geographic regions.

In this paper, we study the spatio-temporal patterns of reported socio-economic drought impacts of both multi-year and single-year drought periods in Germany between 2000 and 2022. Germany is selected as a case study because of its recent history of significant droughts (2003, 2015, 2018-2022, see Fig. A.6 for annual soil moisture observations) with widespread impacts on agriculture, forestry, livestock, and waterways navigation, among others (Peña-Angulo et al., 2022; Rakovec et al., 2022; Tijdeman et al., 2022; de Brito et al., 2020). The assessment of reported impacts supports a

focal point on their human perception. Specifically, we focus on the multi-year drought period between 2018-2022, which is considered a new benchmark in terms of duration and intensity (Rakovec et al., 2022). With this, we aim to understand (a) how single-year and multi-year drought events differ, (b) how drought impact patterns change or persist over the years, and (c) how land-cover is related to these impact patterns. While a multitude of factors influence the exposure and

vulnerability to drought hazards (Meza et al., 2019; De Stefano et al., 2015; Hedayat and Kaboli, 2024), we focus on land-cover due to its widespread usage, allowing us to demonstrate the analytical capabilities of our approach.

## 2. Methods

In this study, we used newspaper texts to create a drought impact dataset covering multiple sectors in Germany. We introduced the concept of a drought impact profile (DIP) to construct a typology summarizing co-occurring drought impact types at a certain time and region. Based on the developed DIPs, we investigated patterns of drought impact occurrence throughout 3 levels of analysis, which provide increasing depth to understanding the characteristics of multi-year droughts (see Fig. 1). First, we compared the DIPs of multiple drought events to understand how single-year and multi-year drought events differ. Second, we examined how the DIPs change or persist within each district using graphical and sequence mining methods. Third, we used land-cover data to demonstrate how external data on exposure and vulnerability can be linked to the DIPs to understand what controls their occurrences.

### 2.1 Data

We developed a dataset covering 7 commonly observed drought impact types in Germany between 2000 and 2022. These include impacts on agriculture (including crop yield losses), livestock (i.e. impacts on livestock farming and animal populations), waterways (i.e. impacts on shipping and navigation), forestry (i.e. impacts on trees and forest ecosystems), aquaculture (i.e. impacts on fishing-related activities), social (i.e. impacts on places and activities used for recreation, tourism, leisure), and fire (i.e. fire in forests or other areas due to drought conditions); for a detailed description of each impact class, see Table A.1.

To create this dataset, we leveraged the text-mining approach proposed by Sodoge et al. (2023) for detecting and classifying the drought impacts and their geographic location from newspaper articles. We considered ~50,000 German newspaper articles mentioning drought-related keywords published between 2000 and 2022. From this sample of newspaper articles, we first removed duplicate and non-relevant articles. Then, we classified whether each article reports on an impact type using lasso logistic regression models that were trained and evaluated on a sample of 1,800 manually annotated newspaper articles (de Brito et al., 2020). The models achieved an 89% median accuracy when compared to the manually annotated data (see Table A.2). In a final step, we estimated the impact location on the district level following the nomenclature of territorial units for statistics (NUTS-3 units). To this end, we considered the locations mentioned within each text and determined the impacted area based on regions where multiple detected locations cluster. The resulting dataset contains observations describing the frequency of drought impact statements (DIS) by year and district. A DIS documents a specific type of reported impact, its estimated date of occurrence, and its location. For example, a DIS could describe the reported impact on agriculture in Leipzig on 16.8.2022. We aggregated the DIS per year and district. The aggregation by year followed natural breakpoints, as shown in Fig. A.2. Most impacts were reported in summer and continuously decreased towards winter. To assess how well our DIS dataset corresponds to external data, Sodoge et al. (2023) correlated it against multiple empirical indicators: precipitation deficit (DWD, 2023), Google trends data reflecting public awareness (Google, 2023), forest fire statistics (BZL, 2020), and agricultural yield losses (The Regional Database Germany, 2023). The validation results showed that the DIS and these empirical indicators were

correlated, suggesting that our estimates are accurate (see Fig. A.1). Detailed descriptions of the proposed method, validation procedure, and results can be found in Sodoge et al. (2023).

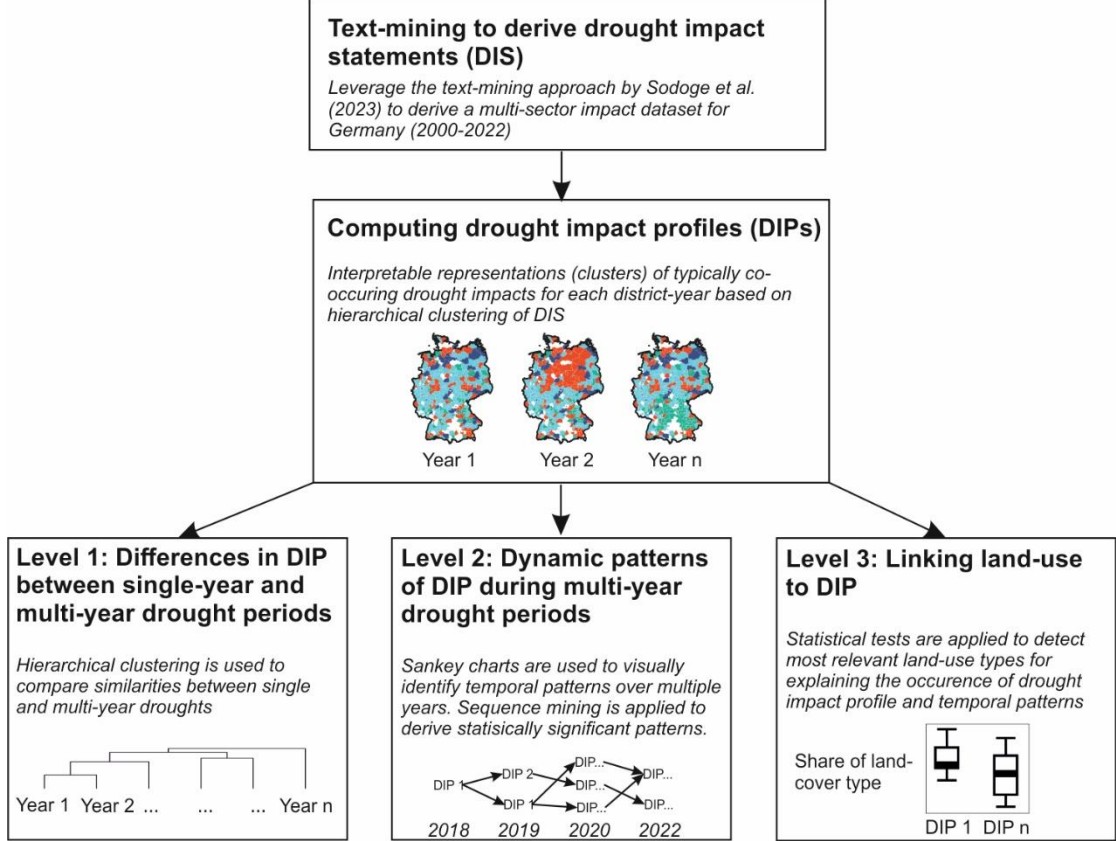

**Figure 1: Overview of the methods used to compute drought impact profiles and analyze their patterns.**

## 2.2 Computing and analyzing drought impact profiles (DIPs)

Drought and their impacts are known to be power-law distributed (Zscheischler et al., 2014; Mahecha et al., 2017). Accordingly, we find that most observations in our DIS database have few or no impacts reported, while a few observations contained the majority of reported impacts. This intrinsic imbalance hinders the construction of DIPs because they disrupt clustering by co-occurrence patterns. Hence, we used the following transformations to the DIS data to ensure that the resulting DIPs primarily reflect co-occurrence patterns rather than the severity of droughts. To this end, we only consider observations from years with observed drought conditions and socio-economic impacts. We selected these based on three criteria: (i) the number of DIS reported per year, (ii) previous research that studied drought events in Germany (Peña-Angulo et al., 2022; Tijdeman et al., 2022; Rakovec et al., 2022; Boeing et al., 2022; Erfurt et al., 2020), and (iii) expert knowledge of hydrologists studying droughts in Germany. Consequently, we selected the droughts of 2003, 2015, and the 2018-2022 multi-year drought period. Along the three criteria, we found an overlap for all selected years. Previous research has focused on these periods, showing relevant observations concerning soil moisture and precipitation deficits. We excluded 2021 from the analysis because few impacts were reported, which could skew the computation of DIPs. Still, we did not treat 2022 as a single-year drought but instead termed 2018-2022 a multi-year drought. We decided to group these years because impacts observed in sectors such as forestry, agriculture, or social in

2022 were still connected to the bio-physical impacts of prior years, which makes separation and independent treatment of 2018-2020 and 2022 difficult. We then grouped the DIS data by year and district and re-scaled from 0 to 1, where 0 means the minimum DIS value within the grouping, and 1 is the maximum (see Fig A.3). This rescaling allowed us to assess variations in the relative significance of impacts across different regions (e.g. north vs. south) and years.

After transforming the data, we created the DIPs by clustering similar observations. To this end, we computed the Euclidean distances (Eq. 1) between each pair of observations (x and y) based on all 7 impact types ($imp_{1-7}$). A small distance reflects similar impacts, whereas a larger distance indicates distinct ones. Based on these distances, we clustered the observations using an agglomerative hierarchical clustering algorithm called Ward's linkage (Sharma et al., 2019). We selected this algorithm because it is known to provide robust results when dealing with continuous data by minimizing the variance between clusters. The disadvantages of the specific clustering algorithm are its sensitivity to outliers and its tendency to form equal-sized clusters (Sharma et al., 2019). This method initially labels each observation as an individual cluster and iteratively merges them into larger clusters based on the identified distances (Husson et al., 2010). The ideal number of clusters (k) was determined using both quantitative measures (e.g. Elbow method, silhouette coefficient) (see Ketchen and Shook, 1996; Thorndike, 1953; Zambelli, 2016) and domain knowledge. For the latter, we considered existing information about compounding and cascading impacts in Germany (de Brito, 2021) as well as co-occurrence patterns within our DIS dataset (Fig. A..4).

$$d(x,y) = (x_{imp1} - y_{imp1})^2 + \cdots + (x_{imp7} - y_{imp7})^2 \tag{1}$$

### 2.2.1 Differences in impact patterns between single-year and multi-year drought events

To compare the impact patterns of single-year and multi-year drought events, (analysis level 1 in Fig. 1), we used a similarity measure and hierarchical clustering. We computed the similarity between the two years by counting the number of districts with identical DIP in both years. For instance, if 140 districts exhibited the same DIP in 2003 and 2015, the similarity measure would also be 140. Subsequently, we applied hierarchical clustering with Ward's linkage to visualize these pairwise similarities in a dendrogram.

To further explore the differences between single-year and multi-year droughts, we also considered the diversity of occurring DIPs. To this end, we calculated the Shannon index (H) (Spellerberg and Fedor, 2003) for each year by summing the products of the relative abundance (p) of each category (i) and the natural logarithm of that category's relative abundance (ln(pi)) (see Eq. 2). A higher H value suggests that there are many different types of DIPs across the analyzed districts, and these are evenly distributed. In contrast, a lower H value indicates fewer distinct types of DIPs, and some may dominate.

$$H = -\sum_i p_i * \ln(p_i) \tag{2}$$

### 2.2.2 Dynamic patterns of impacts during multi-year drought periods

To investigate how the DIP patterns evolved during the 2018-2022 multi-year drought period (analysis level 2 in Fig. 1), we employed two distinct yet complementary approaches: a graphical analysis using alluvial diagrams and statistical

sequence mining. Both approaches aimed at identifying temporal sequences that describe DIP's characteristic shifts (or persistence). Alluvial charts served to effectively visualize sequences, presenting them in proportion to the number of affected districts. Sequence mining assisted as a quantitatively complementary approach to identify statistically significant sequences of DIPs in consecutive years. We employed the CSPADE (Sequential Pattern Discovery using Equivalence classes) algorithm, a widely used sequence mining implementation (Zaki, 2001; Wright et al., 2015) (see Fig. 2). To apply the CSPADE algorithm, we created a transactional database with the antecedent and consequent DIPs in each district during the 2018-2022 drought period. The extracted sequences defined by two connected elements (a and b) were evaluated on 3 measures: support, confidence, and lift. Support corresponds to how often the particular sequence appeared within the data (see Eq. 3). Confidence measures how often the DIP occurred together relative to all observations with the antecedent (see Eq. 4). Lift measures how often antecedent and consequent DIPs were observed together relative to how often they were expected to be observed (see Eq. 5). The obtained sequences with high lift can be interpreted as the most prevalent ones.

$$support(a) = \frac{number\ of\ sequences\ containing\ pattern\ a}{total\ number\ of\ sequences} \qquad (3)$$

$$confidence\ (a \rightarrow b) = \frac{support(a+b)}{support\ (a)} \qquad (4)$$

$$lift\ (a \rightarrow b) = \frac{support\ (a+b)}{support\ (a)*Support\ (b)} \qquad (5)$$

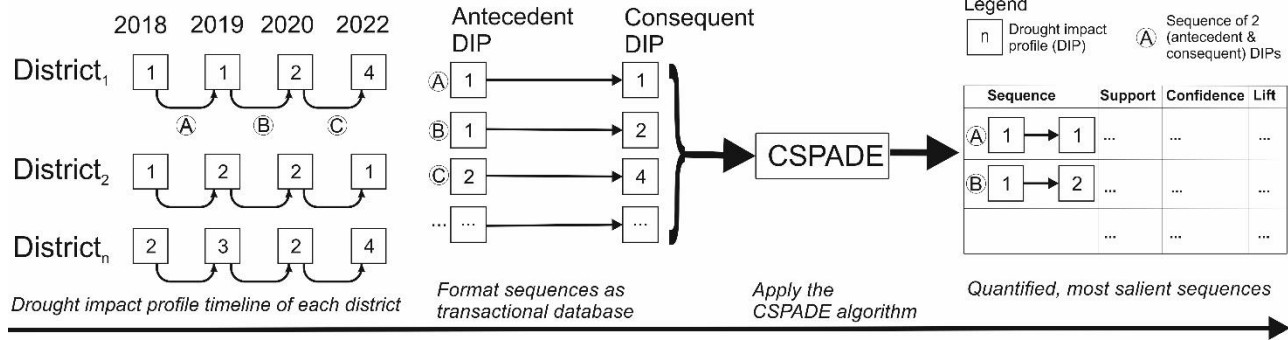

**Figure 2: Process of extracting DIP sequences with CSPADE algorithm. For each district, a timeline was created representing the DIP observed during the multi-year drought period. These were split into a transactional database representing the transitions between the DIPs of two consecutive years. By using the CSPADE algorithm, we extracted the most frequent sequences and quantified them based on lift, support, and confidence metrics.**

### 2.2.3 Linking land-cover to multi-year drought impact profiles (DIPs)

To demonstrate the analytical capabilities of the DIPs typology, we leveraged non-parametric statistical tests to search for significant associations between the DIPs and land-cover data (analysis level 3 in Fig. 1). For this analysis, we focused on the observations during the multi-year drought period as we aim to disentangle the patterns within this special period. Land-cover has been found to control the effect of drought on ecology (Flach et al., 2021) and the socio-economic impacts of drought (Sutanto et al., 2019; Blauhut et al., 2016). We considered the 10 most prevalent types of land-cover in Germany using the CORINE (Coordination of Information on the Environment, Land Cover) dataset (Büttner et al., 2004). For each district, we calculated the relative share of each land-cover type. To detect statistically significant

associations between the DIPs and the share of each land-cover type, we applied one-sided Mann-Whitney U hypotheses tests. This test was chosen because of its suitability in detecting significant differences between two data samples without requiring specific data distributions by using rank transformation and comparison (details can be found in McKnight and Najab (2010)). For this study, we performed two types of hypothesis testing. First, we tested whether the relative share of a specific land-cover type was higher in districts affected than those unaffected by a particular DIP, which would be indicated by a significant p-value. Second, we compared the land-covers of districts experiencing a particular DIP sequence to those without that sequence. Here, we select the most prevalent sequences from the sequence mining application (see Section 2.2.2). Using sequences can provide insights into what factors drive regions to switch from one DIP to another.

## 3. Results

### 3.1 Socio-economic drought impact dataset

The text-mining-based drought impact dataset for Germany comprises 31,370 DIS along 7 impact types reported by newspapers between 2000 and 2022 (see Fig. 3). Notably, the period from 2018-2022 (excluding 2021) accounts for 42 % of all DIS. Throughout this period, we observe a varied and diverse distribution of the DIS across time and space. For example, northeastern Germany's agriculture and livestock sectors were particularly affected. Conversely, impact types such as 'social', 'forestry', or 'fire' exhibit a more widespread occurrence. The selected drought events of 2003 and 2018 have caused the highest number of impacts across all the impact types we analyzed. However, there are variations in their temporal trends. For instance, 'agriculture' impacts peaked during the 2018 drought. Instead, 'forestry' impacts were less pronounced in 2018 and peaked in 2019 and 2020. Despite the widespread distribution of the impacts, we consistently found dominant drought impacts in northeastern Germany. For instance, the federal states of Brandenburg, Saxony, and Mecklenburg-Vorpommern all located in the east contained the districts with the most impacts reported in agriculture, livestock, and aquaculture.

### 3.2 Drought impact profiles

As a result of the hierarchical clustering, we identified 4 clusters of observations with similar co-occurring impact types, referred to here as drought impact profiles (DIPs) (Fig. 4). Overall, both quantitative evaluation metrics (i.e. silhouette coefficient and dendrogram inspection) and qualitative inspection of the DIPs, confirm the distinctiveness of these 4 clusters (see Fig. A.5). With an emphasis on interpretability, the derived DIPs showcase unique characteristics which closely mirror co-occurrence patterns from correlation analysis results (see Fig. A.4). The silhouette coefficient, measuring 0.22, suggests a moderate degree of separation and discernible structure within the data. In light of the exploratory nature of this study, the moderate clustering results can be considered suitable as they uphold interpretability and align with domain knowledge.

Each DIP is enriched by characteristic impact types and has a varying spatial and temporal distribution. These are used here as a reference point for subsequent analysis. For example, DIP 1 predominantly features 'agriculture' and 'livestock' impacts and is particularly prevalent in eastern Germany. The prevalence of DIP 1 declined during the multi-year drought period. Meanwhile, it was dominant during 2018 and 2003. The second DIP is enriched by water ecosystem consequences, including 'waterways' and 'aquaculture' impacts. In 2018 and 2022, DIP 2 reached its peak when

230 compound heat and drought events affected aquaculture and led to low flows, limiting waterway transportation on major water courses (Conradt et al., 2023; Free et al., 2023). As such, DIP 2 is prevalent in districts with major water courses, such as the Rhine River in western Germany and the Oder River on the Polish border. DIP 3, on the other hand, is composed mainly of 'forestry' impacts and is spread across Germany, especially in forestry ecosystems that experienced notable drought effects: the Harz region, Saxon Switzerland mountains, and Alsace (Holzwarth et al., 2020; Erfurt et al.,

235 2020). While DIP 3 hardly occurred during the single-year drought events, we observed an increase in 2019. Lastly, DIP 4 is characterized by the interplay between 'fire' and 'social' impacts. The occurrence of forest fires, or a high likelihood of them, limits the functioning of recreational zones, such as parks and forests. Notably, we observe an increasing occurrence of this DIP over the last 20 years.

240

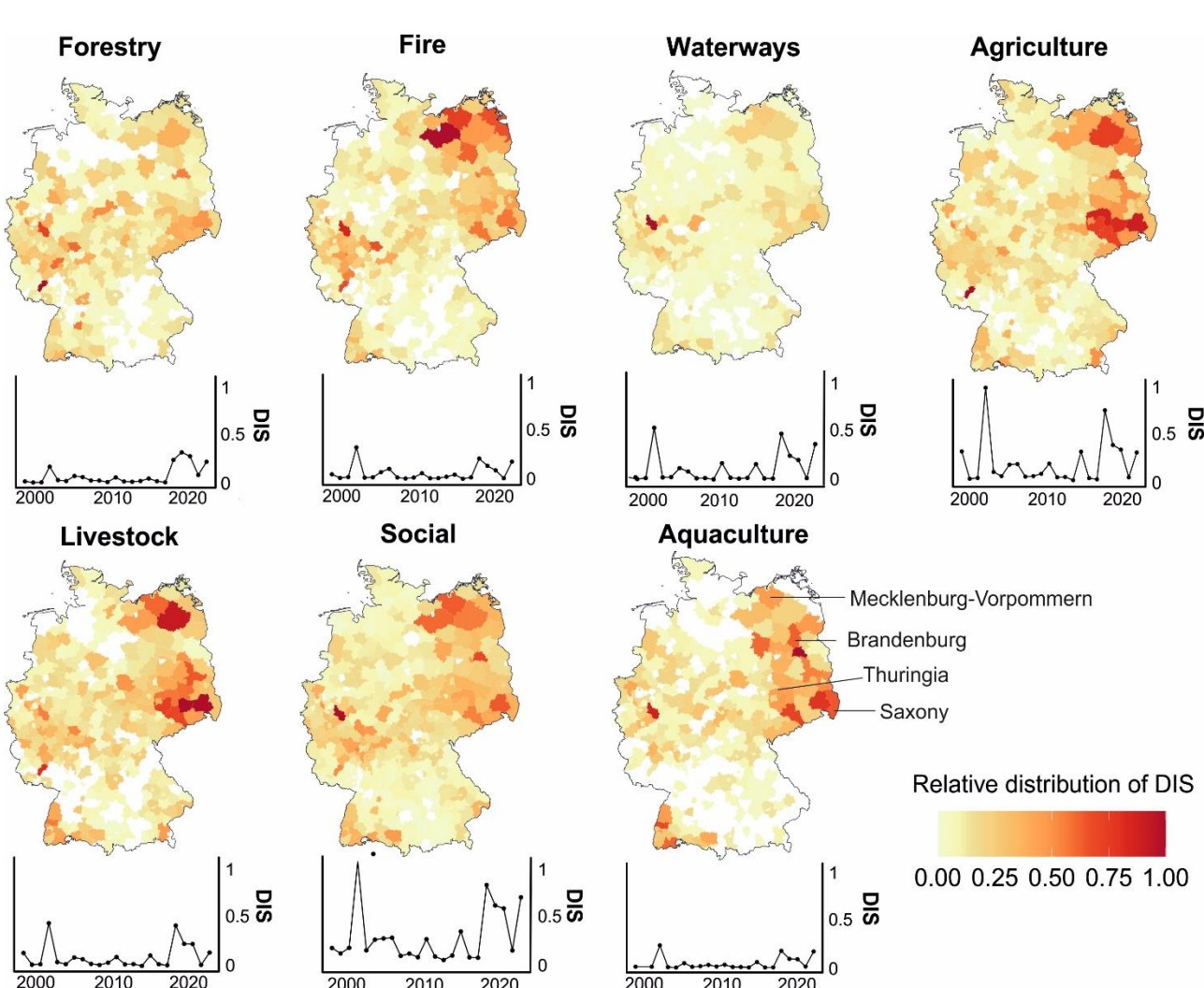

**Figure 3: Spatial and temporal distribution of drought impact statements (DIS) between 2000-2022. Each map displays the relative distribution for a particular impact type, where 1 corresponds to the DIS type's national maximum and 0 to the minimum. Each time series displays the magnitude of DIS, where 1 corresponds to the maximum DIS among all impact types, and 0 is the minimum.**

245

### 3.3 Comparison of the DIPs between single-year and multi-year drought events

The comparison of the DIPs across the drought events shows the distinctiveness of the multi-year drought period compared to prior single-year events (Fig. 5). The droughts of 2003 and 2015 display the highest similarities despite being more than a decade apart. Both share a high prevalence of DPI 1 (enriched in agriculture and livestock impacts), particularly in eastern Germany and many districts without any impact. Both years exhibit the lowest DIP diversity scores, corroborating the hypothesis that single-year droughts tend to have more homogeneous impacts.

The number of districts being affected (thus having a DIP) sets the single-year from the multi-year drought events apart. In single-year droughts, an average of 60% of all districts in Germany were affected, whereas, during the 2018-2022 drought, 92% of the districts had at least one reported impact each year. For instance, while 2018 is similar to 2003 and 2015 concerning dominating DIP 1, the spatial extent of the impacts in 2018 is strongly different. Only in 2020, where 21% of the districts did not report any DIS, did the scores display higher similarity to the single-year events. During the multi-year drought period, the varying similarities between each year indicate some evolving differences. A striking finding is the low similarity between 2018 and 2019.

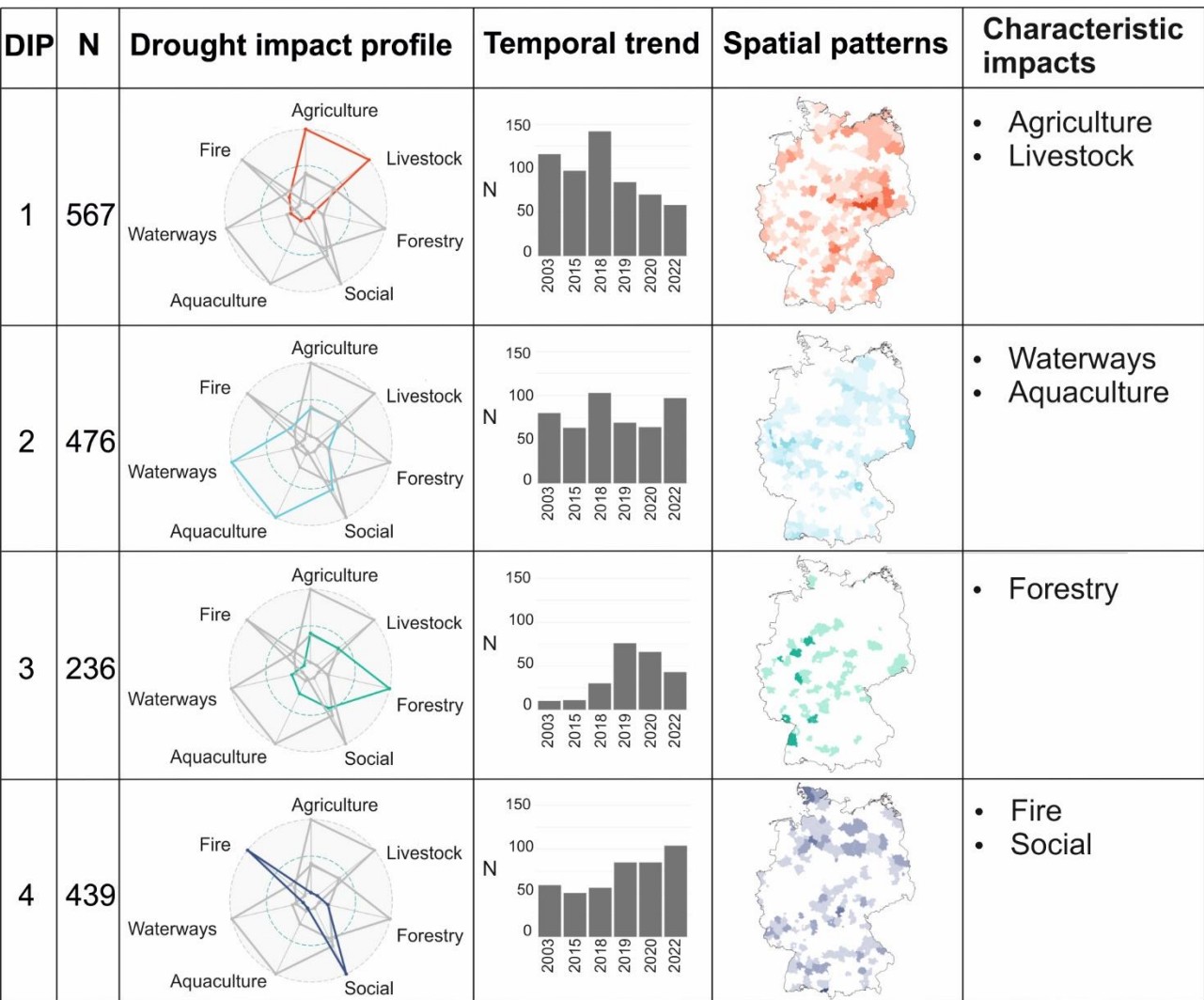

**Figure 4: Overview of drought impact profiles (DIPs) derived from hierarchical clustering (Fig. A.5). Each DIP describes a**

characteristic combination of co-occurrences among the 7 DIS categories. For each impact type in the radar chart, the maximum and minimum correspond to the maximum/minimum of the particular impact type in the DIS dataset. For the spatial patterns, DIPs are aggregated for the analyzed years.

### 3.4 Dynamic patterns of impacts during multi-year drought periods

Throughout the multi-year drought period, we observe distinct patterns of how the DIPs change over time. The probability that a district remains with the identical DIP for two years is only 26 %. Yet, the DIPs do not change in random order and instead follow identifiable patterns that cause shifts in the dominating DIP. By examining the results from both the alluvial chart analysis and sequence mining, we identify 4 major trends (Fig. 6 and 7).

First, we recognize a legacy effect driving a delayed emergence of the 'forestry' DIP from 2019 onwards. From 2018 to 2019, 53 (13%) districts shifted from the 'aquaculture/waterways' and 'livestock/agriculture' DIPs to the 'forestry' DIP. Sequence mining also revealed similar sequences (DIP 1 to 3, support=0.20 in Fig. 6). This underlines the escalating significance of the forestry sector in 2019. After 2019, the prevalence of the 'forestry' DIP slowly declined, yet it remained at higher levels than in 2018.

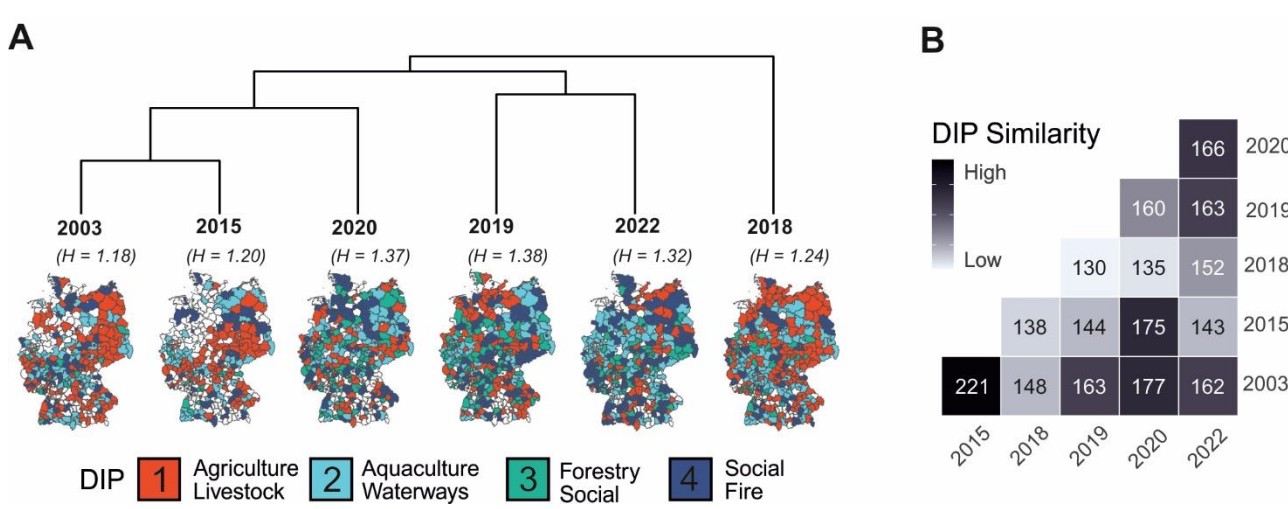

**Figure 5: Comparison of annual events based on DIPs within each district. A dendrogram of hierarchical clustering where a structure of similar years emerges. The height dimension within the dendrogram refers to the dissimilarity between the years. H indicates the calculated diversity index. B similarity matrix with the number of identical DIPs between individual years and is used to perform the hierarchical clustering in panel A.**

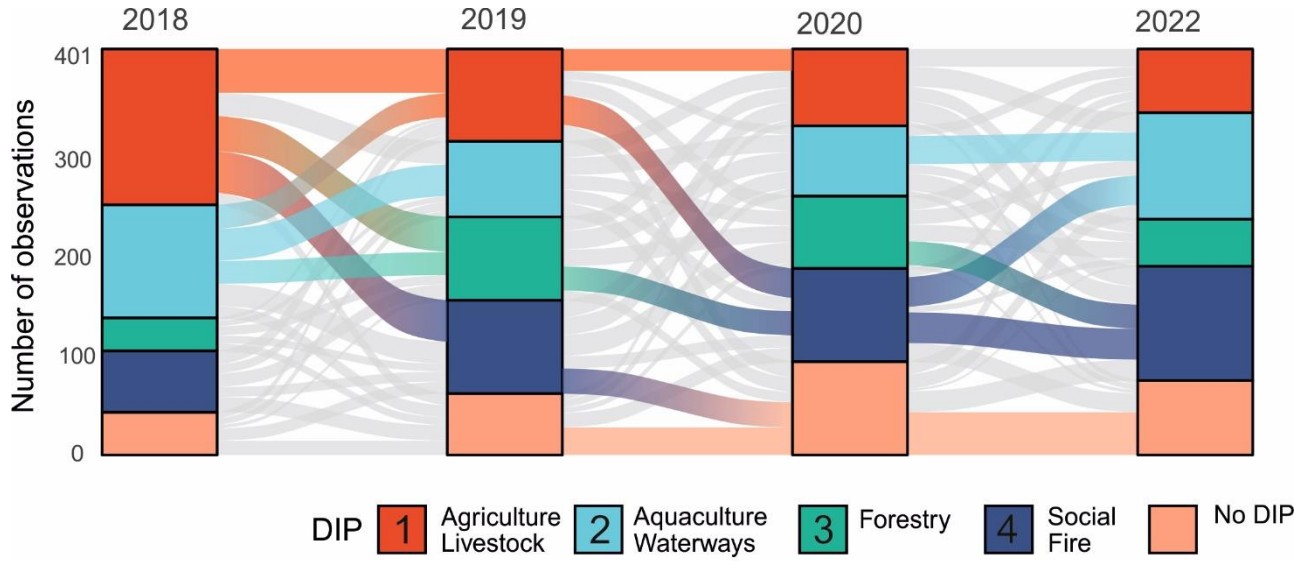

**Figure 6: Transitions among DIPs present in each district during the multi-year drought period. Flows of at least 20 districts between two DIPs are highlighted.**


Second, we identify an increasing prevalence of the 'social/fire' DIP, which was present in 13% of the districts in 2018 and increased to 26% in 2022. Within this context, 65 districts affected by 'agriculture/livestock' DIPs in 2018 and 2019 shifted to 'social/fire' DIPs. Additionally, 44 districts associated with the 'forestry' DIP in 2019 and 2020 shifted to the 'social/fire' DIP in 2022. Here, we hypothesize that severe and long-lasting forest damages reported in the prior 2 years

had resulted in a loss of forest function for recreation or made forests more vulnerable to fire. Then, the shift towards 'social/fire' DIP would directly result from the multiple years of drought that have damaged forest ecosystems.

Third, the prior two trends are underpinned by a steadily decreasing relevance of the 'agriculture/livestock' DIP and a more even distribution of the DIPs in the consecutive years. In 2018, 142 districts were linked to the 'agriculture/livestock' DIP, while in 2022 only 58 were affected. This decreasing relevance results in a more even

representation of the DIPs in the following years, which is visible in the measured DIP diversity (see Fig. 5). Concurrently, a more fragmented geographic distribution of the DIPs emerges. For instance, north-eastern Germany is less dominated by the 'agriculture/livestock' DIP.

Fourth, we found that districts affected by the 'waterways/aquaculture' DIP exhibit a higher degree of persistence, meaning that they are less likely to transition to other DIPs. The sequence mining highlights a sequence where districts

remain with the 'waterways/aquaculture' DIP for two years (DIP 2 → DIP2, support = 0.246 in Fig. 6). This persistence can be attributed to the importance of waterbodies for specific regions, exemplified by the vital role of waterbodies like the Rhine River. Meanwhile, a less frequent sequence was identified where districts shift from the 'waterways/aquaculture' to the 'social/fire' DIP.

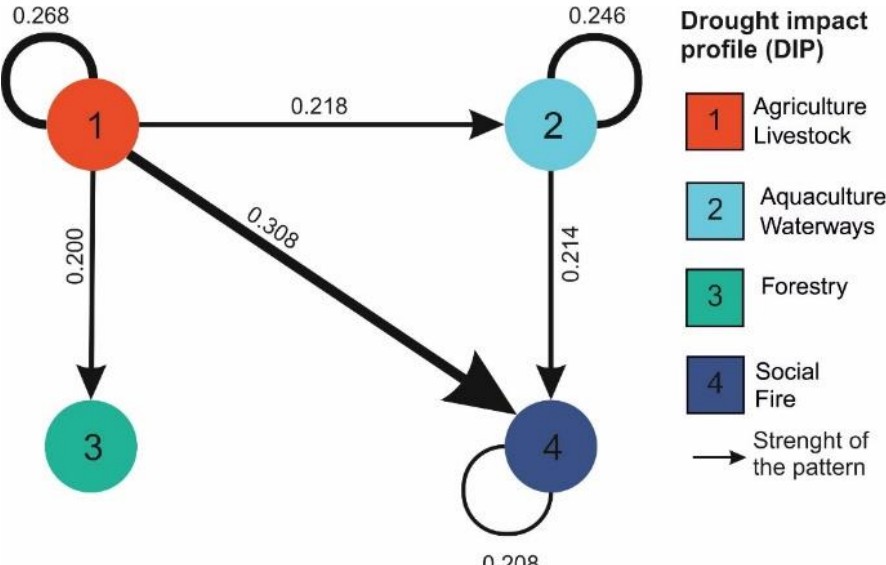

**Figure 7: Sequences of DIPs during multi-year drought period discovered with CSPADE sequence mining algorithm. All sequences with a minimum support measure of 0.2 are displayed and labeled accordingly. Full evaluation metrics are provided in Table A.3.**

### 3.5 Linking land-cover and multi-year drought impact patterns

To investigate the exposure factors contributing to drought impacts, we linked the DIPs with distinct land-cover types (Fig. 8). Our analysis revealed key associations between DIP categories and land-cover types. DIP 1, representing 'agriculture' and 'livestock' impacts, is more prevalent in districts with non-irrigated, arable land than those without this DIP (p-value 0.00). At the same time, districts with agricultural land-cover are more likely to experience 'agriculture' and 'livestock' impacts. DIP 2 ('aquaculture/waterways') is significantly linked to a higher presence of watercourses and

water bodies (p-value = 0.001; 0.003, respectively). Districts impacted by the 'forestry' DIP exhibit elevated levels of broad-leaved and mixed forest land-cover (p-value =0; 0.04), while those influenced by the 'social/fire' DIP show greater proportions of mixed and coniferous forests (p-value =0;0.02). Here, we note a particular differentiation: coniferous forests are significantly linked to the 'fire/social' DIP, whereas broad-leaved forests with the 'forestry' DIP. This distinction points to a higher susceptibility of coniferous forests to 'fire' impacts, while broad-leaved forests appear to be

more affected by factors such as tree mortality. Other significant associations were also found. For example, the 'forestry' DIP is linked to the commercial units' land-cover. While an intuitive linkage cannot explain these findings, these might stem from (i) multi-collinearity among the land-cover types, (ii) unknown characteristics of affected districts or impacts, or (iii) driven by special events. For the outlined example, commercial unit land-cover is highly correlated with urban land-cover, which is more affected by forestry DIP as reports cover tree vitality in an urban context.

To further understand what land-cover types drive districts to shift DIPs from one to another, we identify land-cover types that match the sectors affected by the temporal sequences. For example, districts sticking to the 'forestry' DIP within two consecutive years show significantly higher broad-leaved forest land-cover. This additional analysis adds additional depth to the characteristics of the districts. For instance, districts affected by the 'agriculture/livestock' DIP within two consecutive years display higher shares of agricultural land-cover. Instead, districts that shift from 'agriculture/livestock'

DIP to the 'social/fire' DIP have no significantly higher agricultural land-cover and instead higher coniferous forest land-

cover. These differences indicate that districts remaining impacted by dominating 'agriculture/livestock' impacts possess different land-cover characteristics to those shifting towards other DIPs.

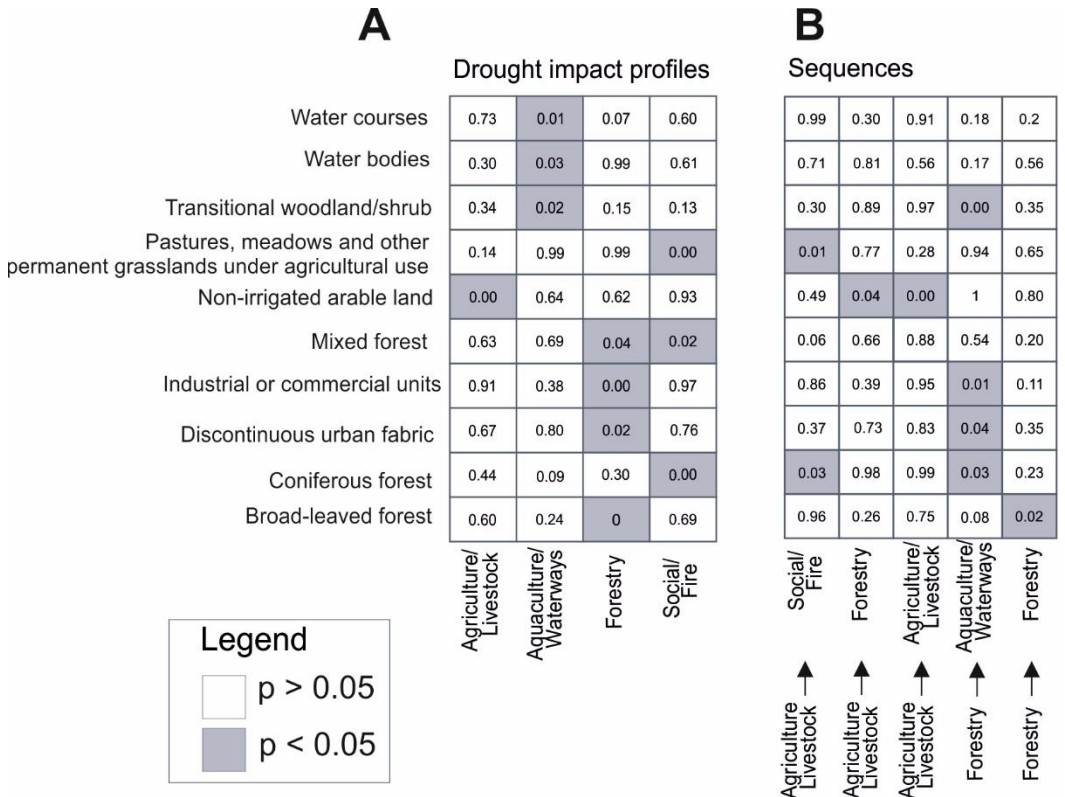

Figure 8: Testing associations between land-cover types and (A) DIP occurrences and (B) DIP sequences according to the one-sided Mann-Whitney U test. A significant p-value indicates that districts where a particular DIP (sequence) indicates have a higher share of respective land-cover types.

## 3. Discussion

Multi-year drought periods are becoming increasingly likely and thus require special attention to develop effective adaptation measures (Rakovec et al., 2022; van der Wiel et al., 2023). Against this background, we investigated the impact patterns during the recent multi-year drought period from 2018-2022 in Germany and compared those with patterns observed in single-year droughts. Using a text-mining-based socio-economic impact dataset, our study provides insights into (1) differences between the multi-year drought and single-year drought events, (2) dynamic patterns during the multi-year drought period, and (3) linkages between land-cover and impact patterns during the multi-year drought period.

Using text-mining to obtain socio-economic drought impact data, we demonstrated how comprehensive drought impact data generated from natural language processing can support the assessment of drought impact patterns. Prior research on drought impact patterns has often been challenged by the lack of multi-sectoral and large-scale impact datasets, which have thus used smaller spatio-temporal scopes. With the increasing availability of impact data generated from newspaper articles and other text data (Noone et al., 2017; Llasat et al., 2009; de Brito et al., 2020; Stahl et al., 2016), scientists can now study drought patterns over long timescales and with broad geographical coverage. The credibility of the derived impact data is highlighted by empirical validations, which demonstrate that the impact's spatial and temporal distribution

correlated with external indicators (Sodoge et al., 2023). Nevertheless, uncertainties need to be acknowledged when using this data type to derive patterns of drought impact occurrence, as described in section 3.1.

Our results illustrate the effectiveness of combining multiple pattern mining methods for examining multi-sectoral and spatio-temporal drought impact patterns, offering both visual and statistical insights. Prior work has used dyadic conceptualization of impact interactions (i.e. the relationships between 2 linked impacts) through network analysis for studying multi-sectoral patterns (de Brito, 2021; Chen et al., 2022). While clustering approaches have already been used to analyze hydrological characteristics of droughts (Kim et al., 2021; Arabzadeh et al., 2016; Hao and Singh, 2015), their use for capturing drought socio-economic impacts remains underexplored. By using unsupervised clustering algorithms, we created a typology of co-occurring impact types (i.e. DIPs) that reflect patterns of cascading and compounding impacts in Germany. Notably, the DIPs display region-specific patterns. For example, the forestry DIP matches relevant forestry ecosystems (Holzwarth et al., 2020), the fire DIP correlates with fire hotspots (Thonfeld et al., 2022), and the waterways DIP aligns with relevant waterways in Germany affected by droughts (Conradt et al., 2023; Free et al., 2023). Thereby, we advanced the representation of multi-sectoral impact patterns. Our approach has the potential to facilitate a multi-sectoral perspective on drought impact patterns as it can incorporate patterns of cascading and compounding impacts. Approaches such as the one applied here have been recently highlighted as valuable tools for investigating the complex patterns of drought impacts (de Brito et al., 2024).

In addition to these methodological contributions, our work also adds to empirical knowledge on droughts in Germany. Concerning the differences between single-year and multi-year drought events, we showed distinct patterns in the multi-year drought event compared to single-year events. The lower spatial extent and diversity of impacts separated the single-year drought events from the multi-year drought period. The widespread impacts of the 2018-2022 drought can be linked to the severe biophysical drought conditions and their extensive reach, which positioned the multi-year drought as an unprecedented event (Rakovec et al., 2022). Agriculture and livestock impacts dominated during the single-year events, while the multi-year drought period displayed a more diverse distribution of impacts. The dominance of 'agriculture' and 'livestock' impacts can be attributed to the importance and vulnerability of the agricultural sector in (northeastern) Germany, as well as the societal significance of the resulting crop yield losses (Zink et al., 2016; Schmitt et al., 2022; Reyer et al., 2012). The high similarity between 2003 and 2015 aligns with soil moisture geographic distributions (Boeing et al., 2022). Specifically for southwestern Germany, Tijdeman et al. (2022) confirmed similar findings for 2003, 2015, 2018, and 2019, which they linked to changing biophysical conditions and the severity of the droughts. For 2003 and 2015, the authors classified both events with the same category titled "intense multi-seasonal drought episodes peaking in summer". Yet, next to the previously identified biophysical differences, our study makes a significant contribution by pointing out the differentiating factors concerning socio-economic impacts.

During the multi-year drought period, we discovered dynamically changing DIPs that led to an increasingly diverse landscape of impacts. In particular, we found that an initial dominance of agriculture/livestock impacts was increasingly replaced by forestry impacts and, subsequently social/fire impacts. The emergence of impacts that increasingly gained relevance during multi-year drought periods reflects evidence from several studies (Tijdeman et al., 2022; Chen et al., 2022; Al-Faraj and Tigkas, 2016). For example, Chen et al. (2022) showed that during a multi-year drought in 1920s China, cascading effects led to unprecedented effects such as growing food prices, dietary changes, and declining health conditions following agricultural losses. Concerning the multi-year drought period under investigation here, particularly the delayed effects on the forestry ecosystem from 2019 onwards, were pointed out by other studies. Repeated stress

exposure caused tree damage that became evident throughout Central Europe (Schuldt et al., 2020; Buras et al., 2020; Kannenberg et al., 2020). Here, we advanced existing knowledge by showing the consequent effects on districts affected in the forestry sector, which later shifted to social impacts as visible in the Harz region (Hahne et al., 2009; Schütte and Plothe, 2022). Next to such sequential patterns, our longitudinal coverage of the multi-year drought period also revealed the sudden effects of extreme events. For instance, the high shares of water-related impacts in 2018 and 2022 were fostered by compounding drought and heat waves (Zscheischler and Fischer, 2020; Wieland and Martinis, 2020). By using a multi-sectoral perspective, we were able to detect such overarching trends that shaped the impact patterns across Germany and connected various sectors.

Our results also demonstrated that distinct land-cover types, such as forest or agricultural land, control the occurrence of impact patterns. We found intuitive connections between land-cover types and the DIPs. For instance, regions with high shares of agricultural land-cover were more likely to experience impacts on agriculture and livestock. We also unveiled subtler effects, demonstrating that coniferous forest land-cover heightened fire-related impacts, which aligns with research findings on German forests (Gnilke and Sanders, 2021). Instead, broad-leaved forests did not exhibit such an association. Identifying factors controlling impact patterns (such as exposure and vulnerability) is necessary to effectively design adaptation measures (Tijdeman et al., 2022; Bachmair et al., 2017; Rannow et al., 2010). Various case studies have demonstrated significant effects of land-cover (and land-use) when assessing drought risk and predicting impacts (Blauhut et al., 2016; Ihinegbu and Ogunwumi, 2022). For instance, Blauhut et al. (2016) found diverse land-cover types relevant for predicting drought risk across Europe. Consequently, the findings on the effect of different land-cover types on impact patterns align with previous research and provide the first insights into underlying mechanisms.

### 3.1 Limitations and future research priorities

Despite the advances presented, it is crucial to acknowledge some limitations in both the data and methods used. Uncertainties exist within the drought impact statements (DIS) data, which can spill over the clustering of the drought impact profiles (DIP). The accuracy of the DIS classification (i.e. the impact type and its location) stems from models with varying uncertainty levels (see Table A.2). While the classification of drought impact types showed high accuracy, recall, and precision levels, fuzzy impact categories such as 'social' proved challenging to classify. While validations of the DIS dataset in Sodoge et al. (2023) showed patterns consistent with external indicators, biases inherent in newspaper coverage, such as missed or overemphasized impacts (Engelmann, 2010), can potentially influence the DIS dataset. Here, we attempted to minimize this by considering a wide range of newspaper outlets (n=260), removing duplicate DIS that report on the same impact and location on the same day, and normalizing the results according to the total volume of newspaper articles per year.

Concerning the methods used, the unsupervised clustering approach employed in our study introduces uncertainties, particularly for districts experiencing impacts that are positioned between two distinct DIPs. In such cases, a slight variation in observed impacts could result in the assignment of a different DIP. Yet, this is a common limitation of all clustering algorithms that seek to identify unique cluster associations as the boundaries are not well-defined (Murtagh and Contreras, 2012). To measure these uncertainties here, we considered quantitative evaluation metrics (i.e. silhouette coefficient and dendrogram inspection) and qualitative inspection of the DIPs. Although we did not conduct sensitivity

analysis tests to quantify these uncertainties (e.g. using soft clustering, see Ferraro and Giordani, 2020)) and their potential
impact on the results, it is important to address these uncertainties.

Concerning the analysis of the effects of land-cover types on drought impact patterns, it is important to acknowledge that the land-cover is only one of many relevant variables relevant to representing exposure and vulnerability. Consequently, interpreting these effects of land-cover needs to consider potential interactions with other factors shown to be relevant in previous research (Meza et al., 2019; De Stefano et al., 2015). For example, irrigation and available adaptation options
in agriculture have been shown to decrease vulnerability (Stephan et al., 2023). Hence, agricultural land-cover alone is not sufficient for explaining mechanisms that drive agricultural impacts. For future research, we suggest investigating the mechanisms more in-depth of how vulnerability factors shape the dynamics of drought impact patterns.

The findings of this study on the dynamic impact patterns during the multi-year drought in Germany provide multiple impulses for future research. First, our findings on the multi-year drought in Germany between 2018 and 2022 require
evaluation against other similar events in this geographic context. Thereby, future research could assess whether our findings are generalizable or constrained to this particular multi-year drought. Second, future research can leverage the identified trends from this research to conduct more in-depth investigations into the mechanisms that underpin these. Such an in-depth, qualitative investigation would act as a complementing counterpart to the macro-level findings discussed here. Third, the links between the identified impact dynamics and meteorological/biophysical processes are a
priority for future research. Since the linkage between meteorological processes and drought impacts is highly non-linear and complex (Sutanto et al., 2019; Shyrokaya et al., 2023), this perspective was not considered within this study, yet it is crucial for better understanding adaptation measure design and impacts under changing climatic conditions.

## 4. Conclusion

In this study, we analyzed the patterns of socio-economic drought impacts during both single-year and multi-year drought
events in Germany. We found that during the multi-year drought period in 2018-2022 in Germany, an increasingly diverse landscape of drought impacts emerged that replaced dominating agriculture and livestock impacts. We noted distinct regional variances in impact patterns, characterized by shifts towards social and forestry-related consequences in some areas and relatively stable agriculture and livestock impacts in others. These findings underscore the need for localized and context-specific approaches to drought management that consider droughts' duration and cumulative effects. Finally,
we demonstrated that these impact patterns are controlled by land-cover types, providing insights into the underlying exposure factors that drive them. Expanding on attributing the impact patterns in future research, we could design more targeted and effective drought adaptation strategies. Overall, our research provides an improved understanding of the unique shifts in socio-economic impacts during the recent multi-year drought period in Germany and highlights the potential of text- and pattern-mining methods to analyze complex drought impact patterns.


**Code availability**: The code for generating the impact dataset is available at https://github.com/jansodoge/drought-impact-text-mining, and the code for the analysis conducted here is provided at https://github.com/jansodoge/drought_impact_profiles_paper


**Data availability**: The newspaper corpus cannot be available due to licensing/copyright reasons. The impact dataset is available at https://github.com/jansodoge/drought_impact_profiles_paper

**Author contribution**:
Jan Sodoge: Conceptualization, Methodology, Software, Investigation, Data Curation, Formal analysis, Writing-Original draft preparation, Writing - Review & Editing, Visualization.
Mariana Madruga de Brito: Conceptualization,  Investigation, Writing - Review & Editing, Visualization, Supervision, Funding acquisition.
Christian Kuhlicke: Conceptualization, Writing - Review & Editing, Project Administration, Supervision, Funding acquisition.
Miguel Mahecha: Methodology, Writing - Review & Editing

**Competing interests**: The authors declare that they have no conflict of interest.

**Acknowledgments:** Uwe Ehret is thanked for his support and consultation in integrating potential information theoretical methods into this research.

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

**Appendix A**


**Table A.1: Definition of impact classes following de Brito et al. (2020)**

| Impact class | Definition |
|---|---|
| Agriculture | Impacts within the agricultural sector including the following sub-categories: reduced productivity of crops, early harvesting, increased need for irrigation, economic losses. |
| Livestock | Impacts within the livestock sector including the following sub-categories: reduced productivity of livestock farming, forced reduction of stock, shortage of feed for livestock, general impacts to animals (including e.g. insect mortality), economic losses for livestock farming |
| Social | Impacts within the social sector including the following sub-categories: parks, tourism, recreation areas and activities affected |
| Forestry | Reduces tree growth or vitality, water stress on trees, decrease in forestry products, increase in pest and disease attacks on trees, increased dieback of trees, economic losses for forestry |
| Aquaculture | Commercial and non-commercial fishing and aquaculture activities |
| Waterways | Impaired navigability of streams (reduction of load, increased need for interim storage transportation of goods at ports) |
| Fire | Occurrence of forest and wildfires |


**Table A.2: Performance of classification models to detect reported drought impacts in newspaper articles**

| Impact class | Recall | Precision | F-score | Accuracy | Sensitivity |
|---|---|---|---|---|---|
| Livestock | 0.92 | 0.93 | 0.93 | 0.88 | 0.92 |
| Fires | 0.97 | 0.95 | 0.96 | 9.93 | 0.97 |
| Forestry | 0.94 | 0.90 | 0.92 | 0.89 | 0.94 |
| Waterways | 0.99 | 0.96 | 0.98 | 0.96 | 0.99 |
| Aquaculture | 0.85 | 0.93 | 0.83 | 0.74 | 0.74 |
| Social | 0.74 | 0.93 | 0.83 | 0.74 | 0.74 |
| Agriculture | 0.92 | 0.94 | 0.93 | 0.89 | 0.92 |


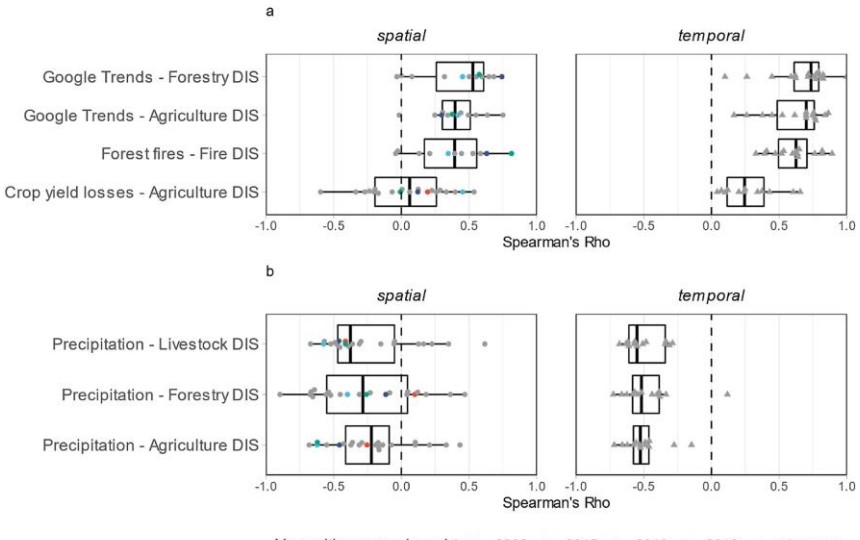

**Figure A.1: Correlation of DIS with external validation indicators from Sodoge et al. (2023). For spatial correlations, each dot represents a year. For temporal correlations, each triangle represents a NUTS-1 unit. Subfigure a) describes correlation analysis in which an ideal explanation corresponds to Spearmans Rho = 1. Subfigure b) describes correlation analysis in which an ideal explanation corresponds to Spearmans Rho = −1.**

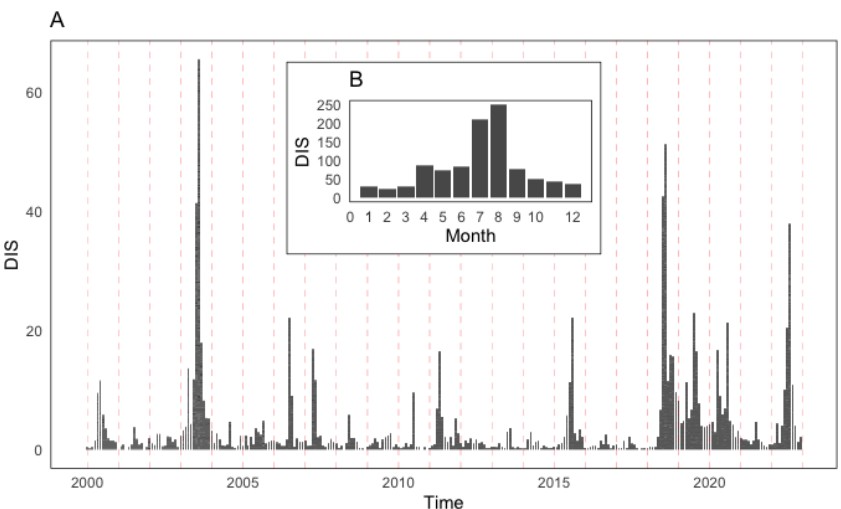

**Figure A.2: Temporal distribution of DIS. (a) Temporal distribution for the entire period studied. Clear peaks exist for studied drought events. (b) total number of DIS per month. A normal distribution with peaks in July/August and only a few impacts reported during winter months.**

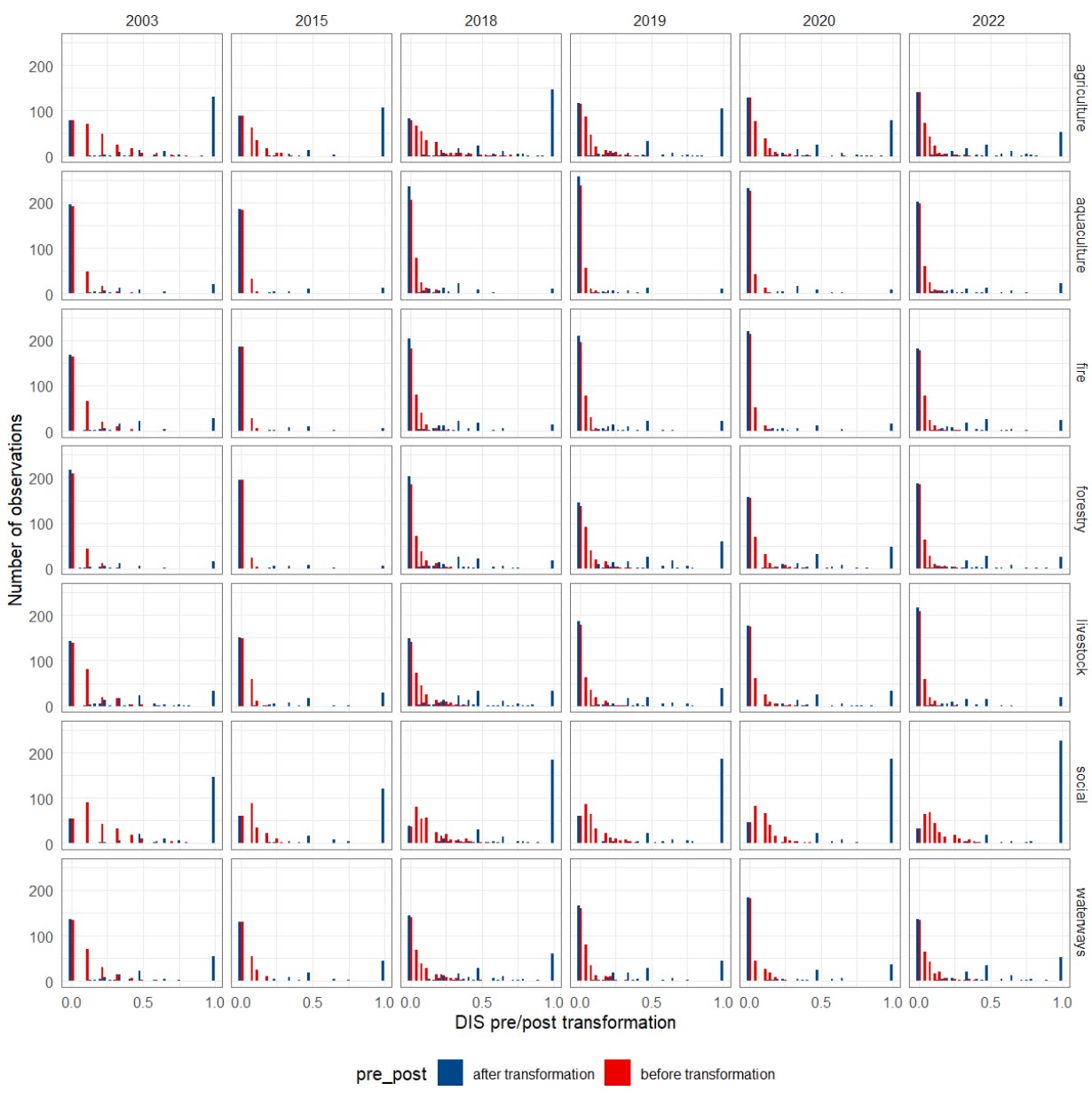

**Figure A.3: Distribution of impacts before and after transformation, re-scaled to [0-1] interval for each grouping.**




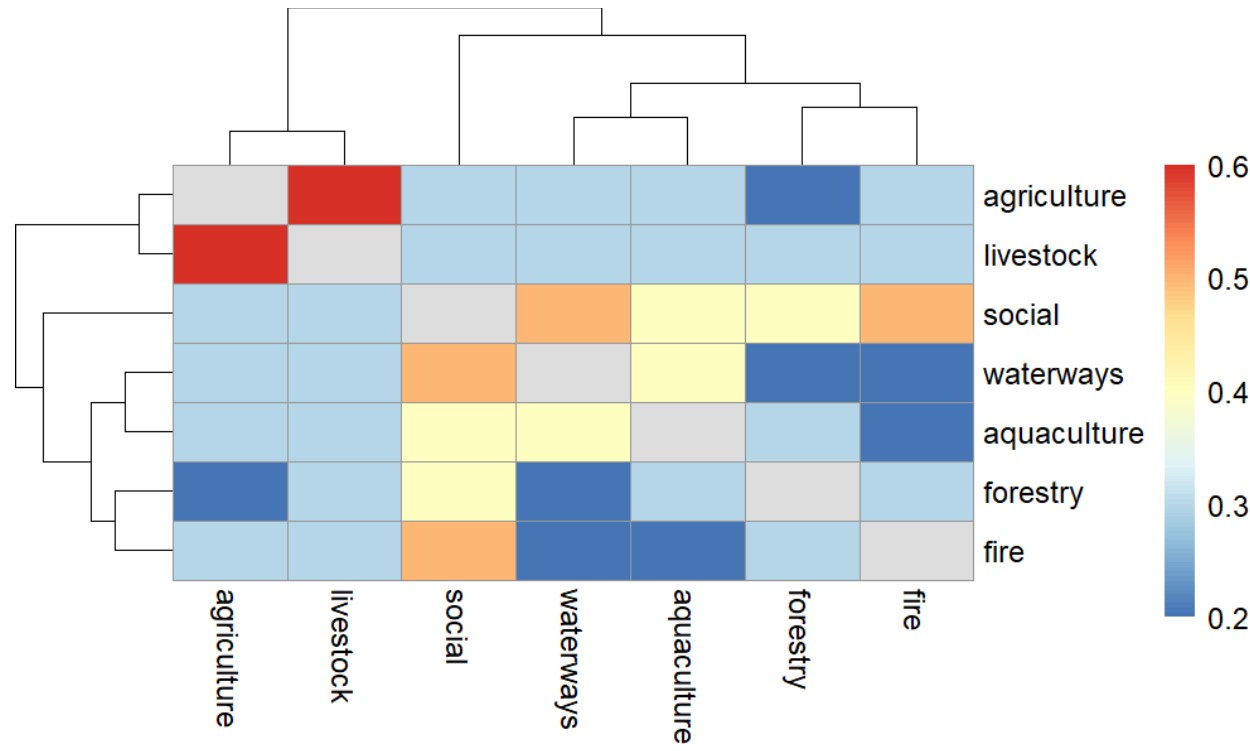


**Figure A.4: Correlations between the occurrences of different impact types. Correlation analysis was performed on the obtained drought impact dataset with annual aggregation before transformation for hierarchical clustering. Correlations are calculated using Spearman's Rho.**


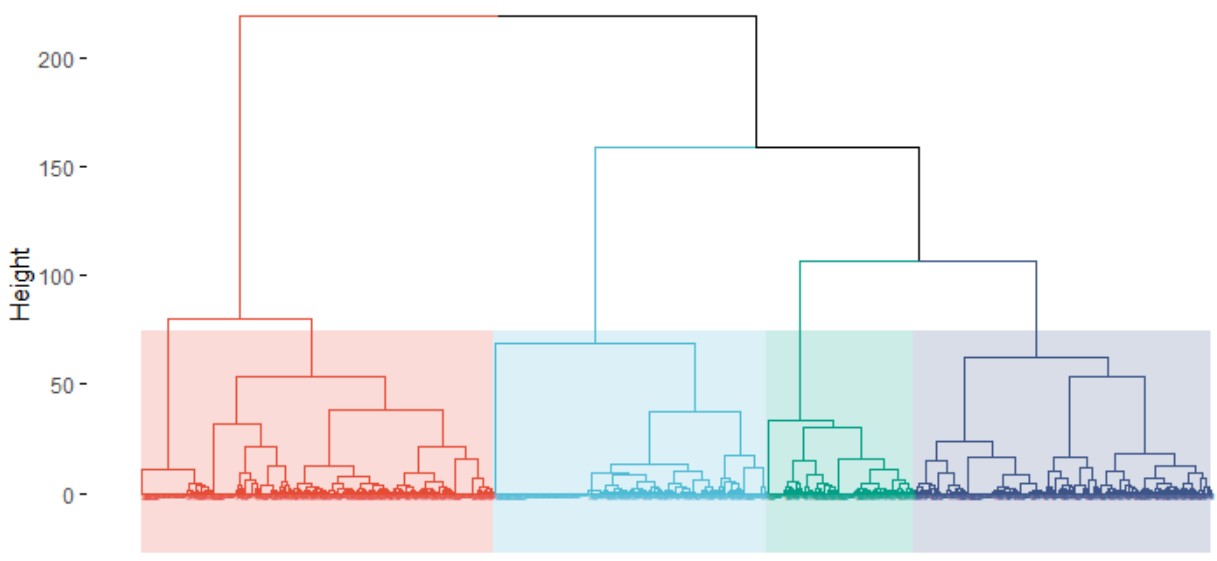

**Figure A.5: Dendrogram of hierarchical clustering of DIS with the 4 clusters colored.**


**Table A.3: Overview of evaluation metrics for obtained sequences**

| Item A | Item B | Support | Confidence | Lift |
|--------|--------|---------|------------|-------|
| 1 | 4 | 0.308 | 0.504 | 0.765 |
| 1 | 1 | 0.268 | 0.438 | 0.717 |
| 2 | 2 | 0.246 | 0.419 | 0.715 |
| 1 | 2 | 0.219 | 0.358 | 0.611 |
| 2 | 4 | 0.214 | 0.364 | 0.552 |
| 4 | 4 | 0.208 | 0.316 | 0.479 |


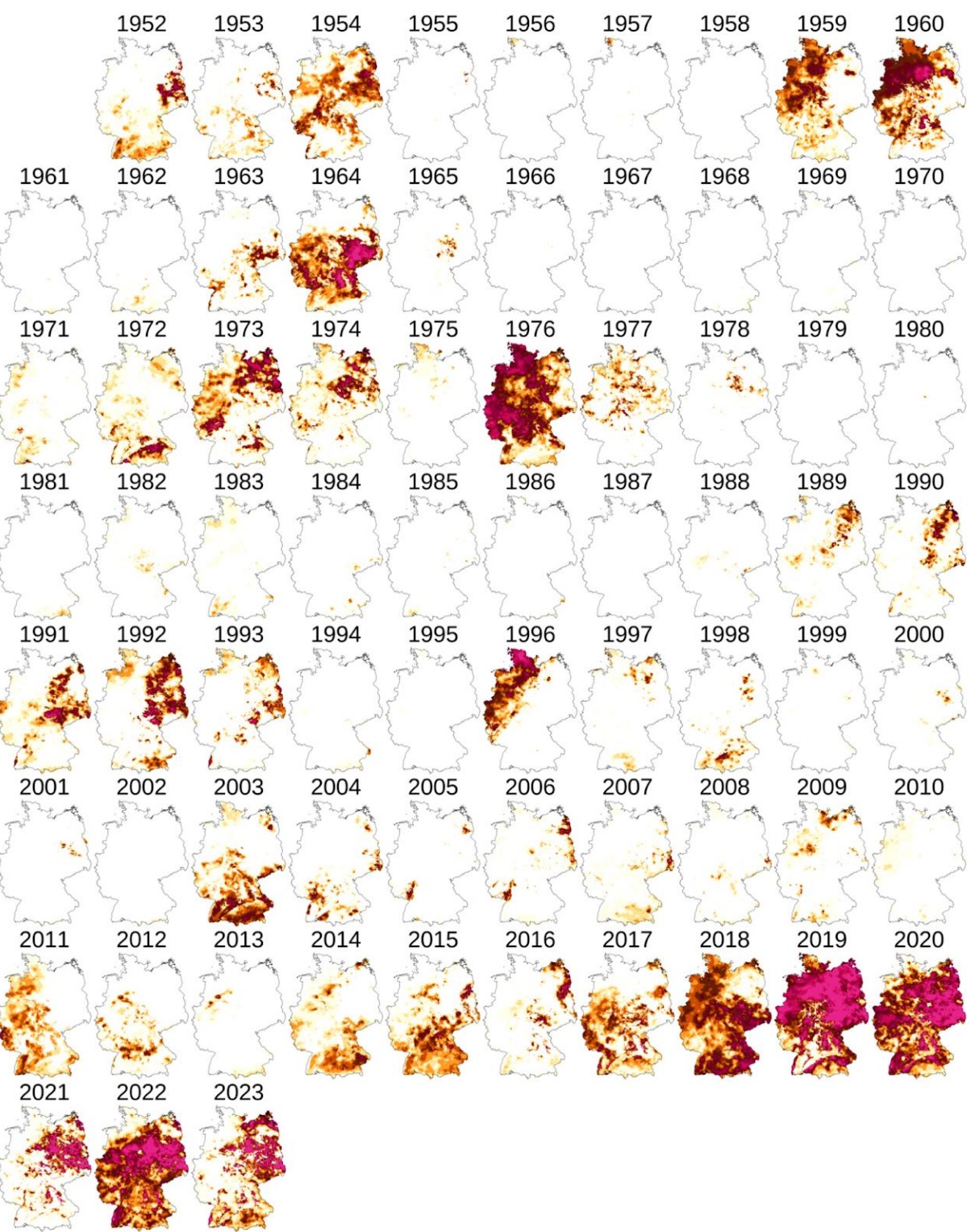

Figure A.6.: *Soil moisture (0-2 meters depth) in Germany, Figure from* Boeing et al. (2022) *and* Zink et al. (2016)*.*