# Peer review of "Text-mining uncovers the unique dynamics of socio-economic impacts of the 2018-2022 multi-year drought in Germany"

_Natural Hazards and Earth System Sciences, 2023_

## Referee Comment (RC2)

Dear Jan Sodoge and co-authors,

we appreciated the opportunity to jointly review your manuscript. Please find our (by Kerstin Stahl and Kathrin Szillat) comments attached below.

Kerstin and Kathrin

**Contribution and General Assessment**

The presented study aims to improve the "understanding of **drought impact dynamics** during increasingly frequent multi-year drought periods". The present work analyzed the patterns of socio-economic drought impacts during two single-year events (2003 and 2015 in Germany) and one multi-year drought event (2018-2022 but excluding 2021). The study used a dataset from media-sourced drought impact statements (DIS) for that analysis.
The main novel contribution (and method) introduced is the DIP (drought impact pattern). The way of analysing drought impact statements is new and we found it generally suitable and interesting for an NHESS readership and really enjoyed reading the paper. The methods appear to be valid, but we do see some need for clarifications. The manuscript is well written and mostly logically presented.  Some sections where we see need for improvements are pointed out in the comments below. Also, we find the conclusions to require some more precision; in particular the title's promise to have uncovered 'unique dynamics' should be toned down and better adjusted to the generalisation that can really be drawn from this one multi-year drought in comparison with (only) two single year events.

**Major Comments**

1) The title is inaccurate. An assertive title claiming to have uncovered (generally) unique dynamics of multi-year droughts (Plural) is a very strong statement that is not supported by the very limited case study and analysis of only one multi-year drought vs two single year events in only one country. This title suggests a much more globally applicable discovery which has yet to be shown. The title has to adapted.

2) The three aims are somewhat different in their levels of analysis depth and their in and interdependence. 3 relates to 1 and 2 and is hence not independent. Also we wonder why only land cover? There are many vulnerability studies around that have shown different sensitivity and adaptive governance aspects besides land cover to matter for drought impacts. The aim must be phrased more as one possible example and needs to be discussed in light of other factors - with suitable acknowledgement of those other controls that have already been shown.

3) The method section was challenging to fully follow for several reasons. It needs improvement to appeal to a wide readership.
a) the description is partly difficult to follow without reading the paper by Sodoge 2023. A bit more on that data would be useful - maybe just an example of a DIS?
b) the structure of the method section is in parts not clean and logic. Some important aspects e.g. in the unnumbered lead text under 2.2 then come up again later etc. This needs considerable editing so that everything is in logical order and only in one place under a clear subheading or clearly introduced paragraph.
c) the DIP clustering is not sufficiently specified. Add equation for the Euclidean distance mentioned somehow in passing in 123. We could not fully follow what exactly went into the Ward algorithm and the elbow method - what exactly is 'height' in Fig. A5?. This algorithm also has some disadvantages that need to be discussed.

4) The correlation with Google Trends: is this really an independent validation dataset? Does it not use the very same media articles that were used to assemble the DIS dataset? If this is the case it cannot not be used for this purpose.

5) The results section contains many different interesting results. These are difficult to appreciate, however, due to a lot of discussion already mixed in. There is a reason why a strong separation between pure results and discussion is generally recommended: it is much easier for readers to see the results of THIS study and hence they will receive much more and more clear appreciation. Interpretations based on literature in particular need to be moved to the discussion section in many places (e.g. lines 132 - 245 contains more discussion than results; or 290ff - either phrase this as a hypothesis to test (methods!) or discuss in the discussion - here it clearly waters down a clear picture of the results and takes the attention off THIS study's achievement).

6) The discussion (and preferably also the analysis, but this may be asking too much) lacks a more quantitative and thorough consideration of the uncertainty, e.g. of the limited sample of events, of the DIS classification (shown with independent data, but are their alternatives to those as well?) and its propagation into the derived DIP patterns? At least a theoretical elaboration, e.g. based on the numbers of correctly/incorrectly classified DIS and the DIP splits or so is needed to stake the limitations. Also the limitation in the base data from Sodoge et al., 2003 need to be considered more here (esp. given that we don't learn so much about it in this manuscript). Overall the discussion section could be more clearly structured and should contain more specific aspects (see comment about moving discussion wrongly placed in results).

**Specific comments**

1. line 23 missing separator (visualization techniques statistical tests)
2. line 62 sounds like those refs only looked at ag. and for., but in fact they looked at more - could be phrased more precisely - overall there is a lot of overcitation with lists of papers the exact contribution of which to the argument is not sufficiently clear - I suggest to weed this out a bit and make the referencing a bit more specific. Otherwise it is not useful and in fact often wrong!
3. line 70 grammatically it should read either "period from 2018 to 2020 "or "period between 2018 and 2020" or "the period of 2018-2020"
4. line 72 land cover is related (singular required)
5. line 75 "Germany-wide"?
6. 75ff three aims and beginning of methods partly redundant - I would find it easier to merge the two at the end of the intro and directly start with 2.1 Data under 2.
7. 86f this is an example of where perhaps a bit more info is required on the data set. Fires are in general mostly ignited by careless people ...a bit more on assumptions/definitions etc. for the reader to get a feeling for the level of determination in the data would be appreciated.
8. 98 'agriculture in Berlin' - in a city? Is this really a convincing example?
9. 132 multi-year drought events? only one event was analysed, so this should be singular. Also see major comment.
10. 140ff define p and i separately and clearly, rel abundance p for each category i?
11. 160 these equations likely do not concur with Copernicus rules for mathematical notation. Define symbols according to those and then use those in equations.
12. 176 Phrasing statistical testing as hypotheses would not only be correct and more clear but also save long somewhat difficult to read text. Suggest to improve.
13. 265ff The placement of the figures is not great for readers and even a bit confusing as Figs 6 and 7 belong together and not Figs 5 and 6.

14. 314 Why is the link not intuitive? There have been so many news on wilting city and park trees in those years that for example these waterbag initiatives started... Urban trees are most affected. There should be literature on it - we suggest to take a look at the actual text of a few DIS. In fact this would be really helpful overall - cite from your data. This is the great advantage of having text data providing the explanation - it doesn't have to be found through correlations with uncertain land cover classification (to put it provocatively) but it is right there to read.
15. 337 This sentence is a bit misleading. The presented work doesn't demonstrate that (you did it in the Sodoge et al. 2023 paper referred to), but how so-derived impacts can be further processed (in theory the analysis could have been done with other impact data as well). The more precise such sentences are the better will be the further use (and) impact of the study!
16. 409 The conclusion is good. But the last few sentences need to be revised with respect to our criticism regarding the title. Also, did the study really demonstrate HOW land cover controls impacts? Or so far mostly 'THAT' with some suggestion for the HOW that lead to the 'more context specific' conclusion?
17. Figures:
18. Fig 1 has very small font sizes and we wonder if it is really needed? The Caption "overview of methods..." suggests a bit more illustration on methods rather than just naming them. At least a conceptual illustration for the three lower boxes as well would be useful then to really see added value (a schematic cluster tree etc.)
19. Fig. 3 suggest to make this a full page (maps are too small)
20. Fig. 4 ID - was this defined as a term?
21. Fig. 8 Font sizes are too small
22. Some minor mistakes in Table A.2 (missing commas)
23. Figure A.2 (peak is in August)
24. The list of references misses doi's. Check the journal's style guide.

---

## Author Comment (AC1)

[Figure]

© UFZ-Dürremonitor/ Helmholtz-Zentrum für Umweltforschung, Friedrich Boeing

*Figure 1: Soil moisture (0-2 meters depth) in Germany, Figure from Boeing et al (2022) and Zink et al (2016). The figure will be added in the updated manuscript to contextualize drought events and add a meteorological perspective*

---

## Author Response (AR1)

*as provided in interactive discussion, added the modified lines in the manuscript*

**Comment 1**: In section 2.2, on the calculation of DIPs, the authors mentioned that in order to prevent errors caused by the data source, only severe drought was chosen as the research object, which is a good consideration. However, the authors didn't explain in more detail in the text how to define severe drought. Is it based on the degree of meteorological drought abnormality or the degree of socio-economic impact? I hope there will be a more detailed explanation.

**Answer**: This is a valid point we will address in the updated manuscript. Similar to your comment on the non-familiarity of readers with the case study, this selection needs to be made clearer. We will modify the text to explain that we used three criteria for selecting the severe droughts: a) the number of drought impact statements i.e. the volume of reported drought impacts in the newspaper database per year, b) previous literature that studied drought events in Germany that indicated that the years of 2003, 2015, 2018, 2019, 2020 were among the most severe during the period for which our newspaper data is available (Rakovec et al. 2022, Erfurt et al. 2019, Boeing et al. 2022), and c) expert knowledge from researchers investigating bio-physical drought conditions in Germany. The literature cited in b) references this assessment to both soil moisture and precipitation deficit. Along these lines, we found an overlap for the selected years, as these were studied in previous research and showed significant values for soil moisture and/or precipitation deficit. While the studies we considered for this selection applied meteorological variables, we did not apply any particular threshold or meteorological condition ourselves. While there is less research on the recent event in 2022, the high volume of reported impacts and low soil moisture led us to include this year, too, despite it lacking more studies compared to the prior years due to its recency. In the manuscript, we will adjust for this by making these selection criteria clearer and using a different terminology than 'selecting severe drought events' which might falsely indicate the usage of some kind of meteorological threshold. Instead, we now refer to this step as "selected drought events".

**Adjusted in manuscript lines:** 120ff

**Comment 2**: Also in section 2.2, the authors mentioned that they deleted 2021 based on "few impacts were reported", but then treated the 2018-2022 drought as a multi-year drought, so why define it that way rather than treating it as a multi-year drought for 2018-2020 with a separate drought in 2022? Does this division affect the subsequent analysis of the changing process of the drought impacts (the analysis in section 3.4 and Figure 6)?

**Answer**: You are indeed raising an important point here, that we will explain in greater detail in the updated manuscript. We decided to stick with the notion of a multi-year drought because impacts observed in sectors such as forestry, agriculture, or social in 2022 were still connected to the bio-physical impacts of prior years, which makes separation and independent treatment of 2018-2020 and 2022 difficult. For example, low soil moisture

persisted in Germany throughout 2021 (see Fig. 1), stemming from the prior years (2018-2020) and thereby driving impacts in 2022, too. Still, due to 801 mm of precipitation in 2021 (586mm in 2018, DWD, 2023), not many DIS were reported compared to 2018, 2019, 2020, and 2022. For instance, in 2018, the reported agricultural impacts were 17 times higher than those observed in 2021. Therefore, considering 2021 would blur the analysis because significantly fewer impacts were reported. Yet, it was important to treat the 2018-2022 period as a multi-year drought period as impacts in 2022 still stemmed from the prior years, too.

We will make sure to improve the explanation as outlined here in the updated manuscript. Also, we will explain this decision as described above not only in the method section but also in the analysis sections you mentioned (3.4 and Figure 6). This does not affect the results in a significant way because the impacts are still connected by mechanisms that we can specify and link to the literature, for instance, the legacy effects on forestry or the social sector. Yet, we will add a remark on the specifics of interpreting these sequences from 2020 to 2022, as you mentioned.

**Adjusted in manuscript lines:** 126ff

**Comment 3**: In addition, based on comment 1) and 2), I would suggest that the authors be able to add a basic overview of the drought situation in Germany in the period 2000-2022, which could help international readers to better understand the context in which this study took place.

**Answer**: This is an important consideration and information that we missed, we will provide this in the updated manuscript. We provide a figure (see Fig. 1 added here as supplement) of annual soil moisture maps for Germany for an extended timeframe based on Boeing et al. (2022) in the supplementary material so the reader can understand the context of the recent drought events also with a historical context. This figure highlights the selected drought events analyzed in this study as well as the severity and uniqueness of the recent multi-year drought period, which is at the center of this research.

**Adjusted in manuscript lines:** 67 and Fig A.6

**Comment 4**: In section 3.1, is it more worthwhile to state and discuss that most DIS are more prominent in eastern Germany than "a varied and diverse distribution of the DIS across time and space" as mentioned in the article? (As shown in Figure 3).

**Answer**: Indeed, we should have pointed this differentiation out more clearly when describing the patterns. While there are few publications explicitly addressing the particular characteristics of Eastern Germany relevant to drought impacts, we will address this feature of the distribution and also provide some first explanations based on the existing literature. The soil moisture decreased more in Eastern Germany compared to other parts of Germany

during the recent drought events (Boeing et al. 2022). Heinrich et al. (2019) highlighted extreme climatic conditions in one of the most severely affected states in Eastern Germany (Mecklenburg-Vorpommern), which recorded 2 °C above the long-term average during the severe drought in 2018. Thonfeld et al. (2022) showed the high spatial concentration of forest damages resulting from pests in Eastern Germany. As these assessments underline the point you made in this comment, this region has a particular role as it is severely affected within different socio-economic sectors and reflected in different bio-physical variables. Thus, it is a relevant feature shaping the picture of socio-economic drought impacts in Germany that should be highlighted accordingly in the updated manuscript and be investigated more closely in the future.

**Adjusted in manuscript lines:** 212
* * *
**Comment 5**: In section 3.3, Lines 230-236, for the discussion of DPI similarity between the 2003 and 2015 droughts, why not try to compare the meteorological drought processes of the two? Perhaps the similar natural anomalies brought about similar impact results?

**Answer**: The similarity of 2003 and 2015 could indeed be a result of similar meteorological processes. As suggested by the soil moisture levels and their geographic distribution for both years (see Fig. 1) there is indeed a similarity between both. While they are distinct from the soil moisture observed during the multi-year drought period regarding severity and spatial extent, they both have similar geographic hotspots with low soil moisture. This can thus partially explain the similarities between these two years. This aligns with the findings by Tijedeman et al. (2022), who classified both events with the same category titled "intense multi-seasonal drought episodes peaking in summer". We will add this explanation in the updated manuscript referring to both soil moisture (Boeing et al. 2022) and the referenced additional literature (Tijedeman et al. 2022). Yet, it is also important to recognize that the linkage between meteorological processes and drought impacts is highly non-linear and complex.

**Adjusted in manuscript lines:** 377ff
* * *
**Comment 6**: Based on comment 5), I would like to further point out that when analyzing the drought impact processes(e.g., section 3.3 and 3.4), the authors do not place the impact processes in the corresponding meteorological context. It is dangerous to conduct a case study without a comparison and examination of natural processes. It is recommended that the authors add consideration of natural processes in their analysis. Perhaps it may help to find that the reason for the shift in the drought impacts of the drought from the meteorological aspect.

**Answer**: We agree with your point about considering meteorological processes to understand socio-economic impacts and differences between years. Our manuscript already touches on this by explaining how certain impacts, like lasting effects on forestry, result from specific meteorological conditions. Following your previous comment we will also add the

outlined Figure on soil moisture and the explanations for similarities between 2003 and 2015. It is indeed important to link these potential explanations that prior studies have developed and we will integrate them as suggested in the comment above. Recognizing the complex mix of meteorological, bio-physical, exposure, and vulnerability factors shaping impacts, we focus our paper on the dynamics of socio-economic impacts primarily. For this research, we focused on comparing the socio-economic impacts of the different drought events. We decided on this focus because of the richly available impact data that we obtained and that prior studies such as Tijdeman et al. (2022) conducted a study that already focused on the bio-physical similarities and differences. Still, as commented by you, our updated manuscript will add these potential explanations as suggested in the prior parts of this answer (e.g. soil moisture comparison 2003 and 2015). Since the linkage between meteorological processes and drought impacts is highly non-linear and complex, we want to offer a broad range of potential explanations based on existing literature that reflect meteorological, bio-physical, and socio-economic mechanisms. Generally, we think that attributing the shifts we observe to bio-physical factors is a very complex and large research endeavor that we will likely tackle in a follow-up research.

**Adjusted in manuscript lines:** 377ff, 442ff
* * *
**Comment 7**: There are some typos in the article, for example, Line 258 forestry should be 'forestry', Line 306 p-value is wrongly written, Line 309 has an extra DIP, etc. It is suggested that the authors carefully review the whole article again when revising.

**Answer**: Thank you for reading thoroughly, we will check for typos and grammar carefully in the updated manuscript.

**Adjusted in manuscript lines:** not specified, all text controlled for typos
* * *
**Comment 1**: The title is inaccurate. An assertive title claiming to have uncovered (generally) unique dynamics of multi-year droughts (Plural) is a very strong statement that is not supported by the very limited case study and analysis of only one multi-year drought vs two single year events in only one country. This title suggests a much more globally applicable discovery which has yet to be shown. The title has to adapted.

**Answer**: We agree with this criticism and the reasoning since we only cover one particular case study and will adjust respectively. Instead, we suggest a new title that highlights the case study to which the findings are related and constrained: "Text-mining uncovers the dynamics of socio-economic impacts during the 2018-2022 multi-year drought in Germany"

**Adjusted in manuscript lines:** 1ff
* * *
**Comment 2**: "The three aims are somewhat different in their levels of analysis depth and their in and interdependence. 3 relates to 1 and 2 and is hence not independent. Also we wonder why only land cover? There are many vulnerability studies around that have shown different sensitivity and adaptive governance aspects besides land cover to matter for drought impacts. The aim must be phrased more as one possible example and needs to be discussed in light of other factors - with suitable acknowledgment of those other controls that have already been shown."

**Answer**: Concerning the interdependence of the three aims/levels of this research design, we did not intend to design them as independent but rather to guide our analysis with increasing depth to understand the characteristics of the multi-year drought by (i) comparing the DIP observed in the multi-year to the single year events, (ii) understanding the dynamics during the multi-year drought, and (iii) linking these with external variables (here land-cover). In the updated manuscript we will adjust these research goals by highlighting that we aim to provide an increasingly in-depth analysis of the multi-year drought.

Concerning the choice of land cover, we agree with your comment that more variables can be considered relevant for such analysis. As you mention, the selection of land cover is rather one of many possible factors to account for. We will adapt the text to emphasize that this is just an example and many others could have been used instead. We use land cover to demonstrate that this approach can account for external variables. Specifically, we want to show how the identified patterns in levels 1 and 2 can be attributed using statistical tests. This research never attempted to provide a full analysis of vulnerability factors which is beyond its scope. Here, we select land cover as a factor as it is used within many studies (e.g. Blauhut et al. 2016, Stephan et al. 2023, Torelló-Sentelles et al. 2022). In the updated manuscript, we will stress this point again in the discussion by pointing to reviews that cover

the extensive range of vulnerability factors (Meza et al., 2019; De Stefano et al., 2015) and that our analysis only provides a snapshot of these. Also, we will stress that the findings regarding the significant effects of particular land-cover types are not sufficient for understanding the mechanisms leading to increased or decreased effects. This is because interpreting these effects of land cover needs to consider potential interactions with other variables that have already been shown relevant in previous research. For example, irrigation and available adaptation options in agriculture have been shown in similar case studies to decrease vulnerability (Stephan et al., 2023). Hence, agricultural land cover alone is not sufficient to explain mechanisms that drive agricultural impacts. Therefore, the discussion will also be updated in the next manuscript version with references to prior studies that have shown the relevance of other variables in similar case studies (see e.g. Erfurt et al. 2019).

**Adjusted in manuscript lines:** 73ff; 80;399ff;431ff
* * *
**Comment 3**: The method section was challenging to fully follow for several reasons. It needs improvement to appeal to a wide readership. a) the description is partly difficult to follow without reading the paper by Sodoge 2023. A bit more on that data would be useful - maybe just an example of a DIS? b) the structure of the method section is in parts not clean and logic. Some important aspects e.g. in the unnumbered lead text under 2.2 then come up again later etc. This needs considerable editing so that everything is in logical order and only in one place under a clear subheading or clearly introduced paragraph. c) the DIP clustering is not sufficiently specified. Add equation for the Euclidean distance mentioned somehow in passing in 123. We could not fully follow what exactly went into the Ward algorithm and the elbow method - what exactly is 'height' in Fig. A5?. This algorithm also has some disadvantages that need to be discussed.

**Answer**: We thank you for the constructive feedback on improving the structure of the methods section, especially given the interlinked description with the previous publication (Sodoge et al. 2023). We will adjust the parts mentioned for improved clarity. We also added a paragraph describing our DIS dataset in more detail. Concerning the DIP clustering we want to address the open questions already here.

The Euclidian distance between two observations x,y is calculated on the transformed DIS dataset to measure similarity in observed impacts (see formula). The Ward hierarchical clustering algorithm is then provided with the distance matrix containing the distances between all pairs of observations calculated using the above-outlined equation. The elbow method is applied to the resulting clustering structure as one of multiple evaluation tools/metrics to determine the optimal number of clusters. In the updated manuscript we will add the reference of Zambelli (2016) for further information on this method. The 'height' in the dendrogram should have been made clear by us and will be added in the updated manuscript in the respective figure caption to add that the height in the resulting dendrogram represents the dissimilarity or linkage distance between clusters. Similarly, we will also discuss the disadvantages of the Ward algorithm in the updated manuscript. These include particularly sensitivity to outliers and a tendency to form equal-sized clusters.

$$d(x, y) = (x_{\text{agriculture DIS}} - y_{\text{agriculture DIS}})^2 + \ldots + (x_{\text{fire DIS}} - y_{\text{fire DIS}})^2$$

**Adjusted in manuscript lines**: complete method section has been enriched with additional details as suggested
* * *
**Comment 4**: The correlation with Google Trends: is this really an independent validation dataset? Does it not use the very same media articles that were used to assemble the DIS dataset? If this is the case it cannot not be used for this purpose.

**Answer**: We agree with the general outline that likely both Google Trends and our newspaper dataset have some linkages. Yet, we would like to point out several arguments that reason for the consideration of Google trends. First, we would like to point out that the validation with Google trends is part of the previous paper, which was already peer-reviewed (see Sodoge et al. 2023). Therefore, a more detailed discussion of the external indicators can be found there (manuscript line 105). Second, we agree that there is a link between both data. Yet there are significant differences that reasoned our selection of Google trends as an indicator in this previous paper. Google Trends measures the online search activities of users in a geographically and temporally defined scope. The DIS dataset that we use measures the volume of 'unique' reported socio-economic drought impacts. Indeed, increased user interest would lead to increased reporting on drought impacts (and vice versa), yet they measure different aspects. Third, in Sodoge et al. (2022) we selected Google trends as one of multiple validation datasets that suited the purpose of comparing to external indicators. Google Trends was also selected because it is frequently used to monitor drought awareness and the perception of communities (e.g. Kam et al. 2019). The conclusion drawn by Sodoge et al. (2022) that the dataset reflects spatial and temporal patterns of external indicators, was drawn from the fact that across various indicators, we found matching correlations where Google trends are not used as the only indicator. Relating to the comment below on uncertainties in the DIS, we will point out the uncertainties of validating with external indicators and of the general DIS dataset in the updated manuscript however (see comment 6 below for details).

**Adjusted in manuscript lines:** not applicable
* * *
**Comment 5**: The results section contains many different interesting results. These are difficult to appreciate, however, due to a lot of discussion already mixed in. There is a reason why a strong separation between pure results and discussion is generally recommended: it is much easier for readers to see the results of THIS study and hence they will receive much more and more clear appreciation. Interpretations based on literature in particular need to be moved to the discussion section in many places (e.g. lines 132 - 245 contains more discussion than results; or 290ff - either phrase this as a hypothesis to test (methods!) or

discuss in the discussion - here it clearly waters down a clear picture of the results and takes the attention off THIS study's achievement).

**Answer**: After careful review of the results section we agree that the manuscript can improve from a more consistent separation of descriptive results and discussion in the context of previous findings. Hence, the updated manuscript will move the interpretation and discussion of findings from the results section to appropriate positions in the discussion section. Also, we will introduce a section in the discussion on limitations (see comment 6).

**Adjusted in manuscript lines:** sections removed from results and insert in discussion in 371f, 375f, 361f
* * *
**Comment 6**: "The discussion (and preferably also the analysis, but this may be asking too much) lacks a more quantitative and thorough consideration of the uncertainty, e.g. of the limited sample of events, of the DIS classification (shown with independent data, but are their alternatives to those as well?) and its propagation into the derived DIP patterns? At least a theoretical elaboration, e.g. based on the numbers of correctly/incorrectly classified DIS and the DIP splits or so is needed to stake the limitations. Also the limitation in the base data from Sodoge et al., 2003 need to be considered more here (esp. given that we don't learn so much about it in this manuscript). Overall the discussion section could be more clearly structured and should contain more specific aspects (see comment about moving discussion wrongly placed in results)."

**Answer**: Highlighting the uncertainty in this approach is a valuable input that we added to a new section entitled "Limitations and future research priorities". This section will include the issues you mentioned on data limitations, and the clustering approach (see the answers to this comment below). Also, it will contain the already mentioned limitations and future research priorities that are outlined in the current version of the manuscript.

As you mentioned, uncertainty could emerge in the classification of the drought impact statements (DIS) and clustering of the drought impact profiles (DIP). We also add these considerations in the updated manuscript's discussion in the paragraphs for the DIS dataset and the DIP formation.

In the classification of DIS, our models identify relevant impact types with varying levels of uncertainty, as indicated by an F-score (selected this metric given the uneven distribution of the training and test data) range of 0.83 to 0.96. These variations in accuracy for different impact types (e.g., the lowest F1 score for social impacts), will be explicitly addressed in the discussion. Yet, we believe that the assessed accuracy allows for a robust estimation of reported impacts. Assessing the accuracy of locating reported impacts in Sodoge et al. (2022) yielded a result of 94%. Concerning the accuracy of the resulting dataset, we assert confidence in the validation procedure outlined in Sodoge et al. (2022), emphasizing its robustness through validation against various external indicators. Despite this, we recognize your comment on the insecurity arising from newspaper articles as an underlying information source, a concern we critically discuss in the paper (line 341). We will explicitly

address the insecurity surrounding the reflection of spatio-temporal trends within the validation analysis and the broader challenges associated with reliable drought impact assessments from newspaper data (see Engelmann 2010). To highlight the general challenges associated with drought impact datasets we will also add Shyrokaya et al. (2021) as a reference.

Concerning the DIP formation through the hierarchical clustering process, this unsupervised clustering approach indeed displays uncertainties which we will address in the discussion of the DIP formation. Specifically, we will underscore this unsupervised procedure (where no true/false labels exist) and the resulting insecurity in assigning DIPs to districts. This is a common limitation of all clustering algorithms which seek to identify unique cluster associations as the boundaries are not well-defined. While direct quantification of this uncertainty is hindered by the absence of labeled data, we will refer to the silhouette coefficient, measuring 0.22, (l. 203), indicating the uncertain/ambiguous distinctions for some observations when assigning DIPs which add a dimension of uncertainty to the results that are yet difficult to quantify. As you recommended, we will integrate these discussions into relevant sections of the paper, addressing insecurity in both DIP and DIS data discussions and emphasizing the relevance of other vulnerability factors, as you suggested.

**Adjusted in manuscript lines:** 411ff
* * *
Specific comments RC2:

**Comment 1**: line 23 missing separator (visualization techniques statistical tests)

**Adjusted in manuscript lines:** 23
* * *
**Comment 2**: line 62 sounds like those refs only looked at ag. and for., but in fact they looked at more - could be phrased more precisely - overall there is a lot of overcitation with lists of papers the exact contribution of which to the argument is not sufficiently clear - I suggest to weed this out a bit and make the referencing a bit more specific. Otherwise it is not useful and in fact often wrong!

**Answer**: This is a mistake and was indeed wrongly communicated by us. As you mention, the cited references are examples of assessments that consider multiple sectors. We will adjust for this, stressing that these are examples of multi-sectorial assessments. Concerning the criticism of 'over citation', we will check each reference for accuracy in the updated manuscript.

**Adjusted in manuscript lines:** 61ff

**Comment 3**: line 70 grammatically it should read either "period from 2018 to 2020 "or "period between 2018 and 2020" or "the period of 2018-2020"

**Adjusted in manuscript lines:** 70
* * *
**Comment 4**: line 72 land cover is related (singular required)

**Adjusted in manuscript lines:** 70
* * *
**Comment 5**: line 75 "Germany-wide"?

**Adjusted in manuscript lines:** 74
* * *
**Comment 6**: 75ff three aims and beginning of methods partly redundant - I would find it easier to merge the two at the end of the intro and directly start with 2.1 Data under 2.

**Adjusted in manuscript lines: removed redundant parts in** 80ff, 72ff
* * *
**Comment 7**: 86f this is an example of where perhaps a bit more info is required on the data set. Fires are in general mostly ignited by careless people ...a bit more on assumptions/definitions etc. for the reader to get a feeling for the level of determination in the data would be appreciated.

**Answer**: We agree that more precise definitions of the drought impact types make the data more transparent to the reader, which we will add to the updated manuscript and also added here below. The typology is based on the work of de Brito et al. (2020). Articles are classified and models are trained to classify articles that report on the following impacts. For additional details, we will also add a reference to the mentioned paper by de Brito et al. (2020).

- Agriculture: including impacts in agriculture related to reduced productivity of crops, early harvesting or selling of products for animal feed, biomass production, and other purposes, increased need for irrigation, increased costs and/or economic losses for agriculture, need for economic help for farmers from public or private institutions
- Livestock: including reduced productivity of livestock farming (e.g. reduced yields or quality of milk and honey, reduced stock weights), forced reduction of stock by early selling or slaughtering, shortage of feed for livestock and other animals, increased costs and/or economic losses for livestock farming, general impacts to animals

- Social: Parks, tourism, recreation facilities, and activities affected (lack of water, risk of fire and/or trees falling, restrictions and impact to tourists and visitors)
- Aquaculture: reduced fish and shellfish yields, early harvesting for alternate uses, increased water needs, elevated costs, and potential economic support requirements from public or private entities.
- Fire: Occurrence of forest and wildfires
- Forestry: Reduced tree growth and vitality, and/or increased occurrence of water stress indicators in forests and trees located in urban areas, decrease in forestry products (e.g. timber, Christmas trees, mushrooms), Increase in pest and disease attacks on trees, Increased dieback of trees, plants or planted tree seedlings, increased costs and/or economic losses for forestry
- Waterways: impaired navigability of streams (reduction of load, increased need for interim storage transportation of goods at ports or humans)

Specifically concerning the comment on fires, the models are provided with articles documenting the occurrence of forest and wildfires. Whether the fire was in fact 'caused' by a human or 'drought' (let's simplify here) cannot be determined by the text classification models. Also, in this research, we do not focus on the drivers of the impacts.

**Adjusted in manuscript lines:** 706ff
* * *
**Comment 8**: 98 'agriculture in Berlin' - in a city? Is this really a convincing example?

**Adjusted in manuscript lines:** 103
* * *
**Comment 9:** 132 multi-year drought events? only one event was analysed, so this should be singular. Also see major comment

**Answer**: plural is used to refer to both single and multi-year events(s), other parts of the manuscript have adjusted for this now referring to the single multi-year drought event
* * *
**Comment 10:** 140ff define p and i separately and clearly, rel abundance p for each category i?

**Adjusted in manuscript lines:** 154ff
* * *
**Comment 11:** 160 these equations likely do not concur with Copernicus rules for mathematical notation. Define symbols according to those and then use those in equations

**Adjusted in manuscript lines:** 175ff

**Comment 12**: 176 Phrasing statistical testing as hypotheses would not only be correct and more clear but also save long somewhat difficult to read text. Suggest to improve.

**Adjusted in manuscript lines:** 194ff
* * *
**Comment 13:** 265ff The placement of the figures is not great for readers and even a bit confusing as Figs 6 and 7 belong together and not Figs 5 and 6

**Adjusted in manuscript lines:** Placement of figures adjusted
* * *
**Comment 14** : 314 Why is the link not intuitive? There have been so many news on wilting city and park trees in those years that for example these waterbag initiatives started... Urban trees are most affected. There should be literature on it - we suggest to take a look at the actual text of a few DIS. In fact this would be really helpful overall - cite from your data. This is the great advantage of having text data providing the explanation - it doesn't have to be found through correlations with uncertain land cover classification (to put it provocatively) but it is right there to read.

**Answer**: The explanation you suggest could sound very legitimate to us and, indeed, the literature highlights such damages, especially in the context of the urban heat island effect. For example, Haase and Hellwig (2022) investigated the severe effect of drought on particularly urban trees. Also, Rötzer et al. (2021) have shown the effect of consecutive drought years on urban tree species. We added these references to the discussion. For future research, we plan to work closer on the text level using qualitative context analysis. Yet we refrain from selectively choosing particular parts of the texts to underline arguments. Instead, for this quantitative analysis, we prefer using the DIS as a numeric measure for the analysis where increased reports link to increased societal relevance. Because of the large number of newspaper articles (approximately 50,000) the spatial and temporal patterns of socio-economic drought impacts emerge rather from the large sample size.

**Adjusted in manuscript lines:** 322ff
* * *
**Comment 15**: 337 This sentence is a bit misleading. The presented work doesn't demonstrate that (you did it in the Sodoge et al. 2023 paper referred to), but how so-derived impacts can be further processed (in theory the analysis could have been done with other impact data as well). The more precise such sentences are the better will be the further use (and) impact of the study!

**Answer**: We agree that this comment is misleading as the demonstration of using text-mining has been achieved in a previous paper. Instead, in the updated manuscript we will rephrase this to highlight how text-generated data can be used for analysis of dynamic drought impact patterns. We will adjust accordingly in this paragraph. We agree that other datasets could also be used for this purpose and suggested it as future research. Still, we believe it is important to highlight the newspaper/text-mining-based aspect since the thereby generated data allows such analysis given the large spatio-temporal scope and volume of text-based information that can be processed.

**Adjusted in manuscript lines:** 345ff
* * *
**Comment 16**: 409 The conclusion is good. But the last few sentences need to be revised with respect to our criticism regarding the title. Also, did the study really demonstrate HOW land cover controls impacts? Or so far mostly 'THAT' with some suggestion for the HOW that lead to the 'more context specific' conclusion?

**Answer**: We agree that these parts need to be adjusted similarly to the adjusted title. First, we will stress that our findings relate to a particular event (the 2018-2022 multi-year drought in Germany) and are not generalizable. For instance, the sentence "Overall, our research provides an improved understanding of the unique shifts in socioeconomic impacts during a multi-year drought period and highlights the potential of text- and pattern-mining methods to analyze complex drought impact patterns." will be adjusted to highlight that '...provides an improved understanding of the dynamic impact patterns during the 2018-2022 multi-year drought period in Germany". Concerning the integration and control for land-cover variables, we will adjust to highlight that we showed an effect of land-cover on the observed dynamics and that future research, which will include more factors, can contribute to understanding the mechanisms.

**Adjusted in manuscript lines:** 455
* * *
**Comments 17-23:** See respective figures for made adjustments; all suggestions that were implemented
* * *
**Comment 24:** The list of references misses doi's. Check the journal's style guide

**Adjusted in manuscript lines:** 480ff

**References**

Blauhut, Veit, et al. "Estimating drought risk across Europe from reported drought impacts, drought indices, and vulnerability factors." Hydrology and Earth System Sciences 20.7 (2016): 2779-2800.

Boeing, Friedrich, et al. "High-resolution drought simulations and comparison to soil moisture observations in Germany." Hydrology and Earth System Sciences 26.19 (2022): 5137-5161.

de Brito, Mariana Madruga, Christian Kuhlicke, and Andreas Marx. "Near-real-time drought impact assessment: a text mining approach on the 2018/19 drought in Germany." Environmental Research Letters 15.10 (2020): 1040a9.

DWD, Deutscher Wetterdienst.
https://www.dwd.de/DE/leistungen/zeitreihen/zeitreihen.html?nn=480164#buehneTop

Erfurt, Mathilde, Rüdiger Glaser, and Veit Blauhut. "Changing impacts and societal responses to drought in southwestern Germany since 1800." Regional Environmental Change 19.8 (2019): 2311-2323.

Haase, Dagmar, and Rebecca Hellwig. "Effects of heat and drought stress on the health status of six urban street tree species in Leipzig, Germany." Trees, Forests and People 8 (2022): 100252.

Kam, Jonghun, Kimberly Stowers, and Sungyoon Kim. "Monitoring of drought awareness from google trends: a case study of the 2011–17 California drought." Weather, Climate, and Society 11.2 (2019): 419-429.

Rakovec, Oldrich, et al. "The 2018–2020 multi-year drought sets a new benchmark in Europe." Earth's Future 10.3 (2022): e2021EF002394.

Rötzer, T., et al. "Urban tree growth and ecosystem services under extreme drought." Agricultural and Forest Meteorology 308 (2021): 108532.

Shyrokaya, Anastasiya, et al. "Advances and gaps in the science and practice of impact-based forecasting of droughts." Wiley Interdisciplinary Reviews: Water (2023): e1698.

Sodoge, Jan, Christian Kuhlicke, and Mariana Madruga de Brito. "Automatized spatio-temporal detection of drought impacts from newspaper articles using natural language processing and machine learning." Weather and Climate Extremes (2023): 100574.

Stephan, Ruth, Kerstin Stahl, and Carsten F. Dormann. "Drought impact prediction across time and space: limits and potentials of text reports." Environmental Research Letters 18.7 (2023): 074004.

Thonfeld, Frank, et al. "A First Assessment of Canopy Cover Loss in Germany's Forests after the 2018–2020 Drought Years." Remote Sensing 14.3 (2022): 562.

Tijdeman, Erik, et al. "Different drought types and the spatial variability in their hazard, impact, and propagation characteristics." Natural Hazards and Earth System Sciences 22.6 (2022): 2099-2116.

Torelló-Sentelles, H. and Franzke, C. L. E.: Drought impact links to meteorological drought indicators and predictability in Spain, Hydrol. Earth Syst. Sci., 26, 1821–1844, https://doi.org/10.5194/hess-26-1821-2022, 2022.

Zambelli, Antoine E. "A data-driven approach to estimating the number of clusters in hierarchical clustering." F1000Research 5 (2016).